# Corticostriatal control of defense behavior in mice induced by auditory looming cues

Zhong Li[1,5], Jin-Xing Wei[1,5], Guang-Wei Zhang[1], Junxiang J. Huang[1,2], Brian Zingg[1,3], Xiyue Wang[1,3], Huizhong W. Tao [1,4✉] & Li I. Zhang [1,4✉]

Animals exhibit innate defense behaviors in response to approaching threats cued by the dynamics of sensory inputs of various modalities. The underlying neural circuits have been mostly studied in the visual system, but remain unclear for other modalities. Here, by utilizing sounds with increasing (vs. decreasing) loudness to mimic looming (vs. receding) objects, we find that looming sounds elicit stereotypical sequential defensive reactions: freezing followed by flight. Both behaviors require the activity of auditory cortex, in particular the sustained type of responses, but are differentially mediated by corticostriatal projections primarily innervating D2 neurons in the tail of the striatum and corticocollicular projections to the superior colliculus, respectively. The behavioral transition from freezing to flight can be attributed to the differential temporal dynamics of the striatal and collicular neurons in their responses to looming sound stimuli. Our results reveal an essential role of the striatum in the innate defense control.

[1] Zilkha Neurogenetic Institute, Keck School of Medicine, University of Southern California, Los Angeles, CA, USA. [2] Program in Biomedical and Biological Sciences, University of Southern California, Los Angeles, CA, USA. [3] Neuroscience Graduate Program, University of Southern California, Los Angeles, CA, USA. [4] Department of Physiology and Neuroscience, Keck School of Medicine, University of Southern California, Los Angeles, CA, USA. [5] These authors contributed equally: Zhong Li, Jin-Xing Wei. ✉email: htao@usc.edu; liizhang@usc.edu

One critical function of central sensory pathways is to detect threatening and potentially dangerous environmental cues and transform the processed sensory signals into motor commands as to quickly initiate appropriate defense behaviors[1–3]. Defense-related sensory neural circuits, which remain not well-understood in various sensory modalities, have been a topic for extensive research in recent years. In natural environments, the most salient dangers include approaching predators and colliding objects. Such looming threats are often cued by the dynamics of sensory signals in different modalities, and may trigger intense defensive reactions. As a popular model for studying such sensory-evoked defense, visual looming stimuli such as expanding dark disks, which mimic approaching aerial predators, have been widely used in experimental settings for rodent studies. These stimuli can trigger freezing or flight behavior depending on the experimental context[4–11]. It has been shown that superior colliculus (SC) neurons play an important role in mediating looming-induced defense behaviors[4–9]. For example, a SC to the lateral posterior nucleus of the thalamus (LP) pathway has been shown to mediate the visually induced freezing[8,9], while a SC to the periaqueductal gray (PAG) pathway mediates the flight behavior[6]. However, whether this architecture of neural circuits for processing sensory threats and inducing behaviors is shared by among different modalities remains unclear.

Compared to the visual system, innate defense circuits in the auditory system have been less well studied. Previously, auditory-related defense behaviors have been more focused on learned behaviors, e.g., the sound-cued fear conditioning[1,12–14], or responses to a sudden loud sound, e.g. the acoustic startle and flight responses[6,15–17]. The latter are more-or-less reflexive, as although modulated by auditory cortical activity, they do not rely on cortical processing[6,15–17]. In a natural acoustic environment, sounds of approaching predators are characterized by a dynamic increase in the sound intensity. Such looming type of sounds is shown to be able to influence emotion and visual information processing in humans and non-human primates[18–21], suggesting that it may be able to induce fear response-like behaviors.

In the present study, we have designed a battery of sound stimuli with increasing loudness (i.e. crescendo sounds) to mimic looming auditory cues. We found that when presented to mice, looming sounds could trigger innate defense behaviors characterized by a stereotypical freezing followed by flight sequence. We further dissected the underlying neural circuitry mechanisms and found that freezing was mediated by the corticostriatal projection to the tail of the striatum (TS) while flight behavior was dependent on corticofugal inputs to SC. In addition, the transition from freezing to flight could be accounted for by the differential temporal dynamics of TS and SC responses to the looming sounds. Our results have revealed a previously unrecognized role of TS in the innate defense control and highlighted the behavioral importance of top-down corticofugal projections in motor control.

## Results

### Looming sounds trigger innate defense behaviors with a stereotypical sequence.
We first examined the behavioral effects of looming-sound stimuli with an open field test similar to that used in visual looming tests[8–10] (see "Methods"). Auditory stimuli were applied through a speaker from one side of the box (Fig. 1a). After we placed a naïve mouse in the box, during the initial 10 min period for habituation, broadband noise at 20 dB sound pressure level (SPL) was continuously applied as a background sound. In the following test period, when the mouse entered a designated center zone, a train of crescendo sound stimuli were triggered (see Methods). For each crescendo stimulus, the intensity of noise was linearly increased from 20 to 70 dB SPL within 0.4 s, then returned to 20 dB SPL within 5 ms and remained at 20 dB SPL for the following 0.6 s (Fig. 1b). We observed a reliable sequence of defense-like behaviors when the crescendo stimuli were applied: the mouse stopped almost immediately and froze for a few seconds (Fig. 1a, blue dot), following which it turned towards and fled to the nest at a high speed (Fig. 1a, red trace). From the analyzed locomotion speed, we were able to distinguish different phases of the sound-evoked behavior: freezing, escape and nest time (Fig. 1c). A great majority of mice we tested (27/32) exhibited this freezing-escape-nest time sequence (Fig. 1g, see Supplementary Movie 1), while only a few escaped immediately without showing freezing (4/32) or only exhibited freezing (1/32).

We quantified the crescendo-triggered behaviors with measurements of freezing duration ($3.0 \pm 0.3$ s, mean ± s.e.m., $n = 32$ mice), latency for escape ($3.5 \pm 0.4$ s, $n = 31$) and normalized top speed ($1.6 \pm 0.1$, $n = 32$) (Fig. 1h–j, Cres). It is known that the visual looming-induced defense behavior exhibits rapid adaption to repeated representations of looming stimuli[5,9]. We thus tested adaptation by repeating the set of crescendo stimuli for five times with 2–5 min inter-trial intervals. We found that the probabilities of freezing and escape were both significantly reduced over repeated trials (Supplementary Fig. 1a). Taking this adaptation effect into consideration, we only tested and measured the first trial in later experiments.

To understand the relationship between auditory intensity and behavioral magnitude, we applied a range of peak intensities (from 50 to 100 dB SPL) for crescendo stimuli (Supplementary Fig. 1b). The probability of escape increased with increasing peak intensities and saturated at 90 dB SPL, while the probability and duration of freezing peaked at 70 dB SPL and then decreased (Supplementary Fig. 1c, d). These results suggest that as the crescendo stimuli become louder, mice tend to switch the defense strategy from freezing-escape to escape-only. In addition, the latency of escape shortened, the top speed increased and the nest time prolonged with the increasing peak intensity (Supplementary Fig. 1d), indicating that the fear level became higher.

We also tested different rise times for crescendo (Supplementary Fig. 2a). We found that 0.4 s rise time was optimal for inducing freezing, as shown by the maximum probability and duration (Supplementary Fig. 2b, c). On the other hand, modulation of escape by the rise time was at most moderate (Supplementary Fig. 2c). It is worth noting that at the shortest rise time applied, the sound stimulus could be analogous to a noise pulse we used previously to induce flight behavior[17]. These behavioral results indicate that escape behavior depends largely on sound intensity while freezing is affected by both the intensity and rise time of looming sounds. Therefore, mice may adopt different defense strategies based on the dynamics of acoustic stimuli.

### Receding sound stimuli fail to trigger defense behaviors.
To further demonstrate that the freezing-escape behavior was in response to a looming object, we redesigned sound stimuli to simulate a receding object (Fig. 1d–e). For the decrescendo (i.e. receding) stimulus with 20 dB SPL background, the intensity of noise decreased linearly from 70 to 20 dB SPL within 0.4 s, opposite to the crescendo stimulus (Fig. 1e). In a group of mice, we tested both the crescendo and decrescendo stimuli, with the two types of stimuli randomly ordered and presented with an interval of at least 48 h (see "Methods"). In contrast to crescendo stimuli, decrescendo sounds failed to induce a reliable freezing or escape behavior (Fig. 1f–g), as shown by the reduced freezing

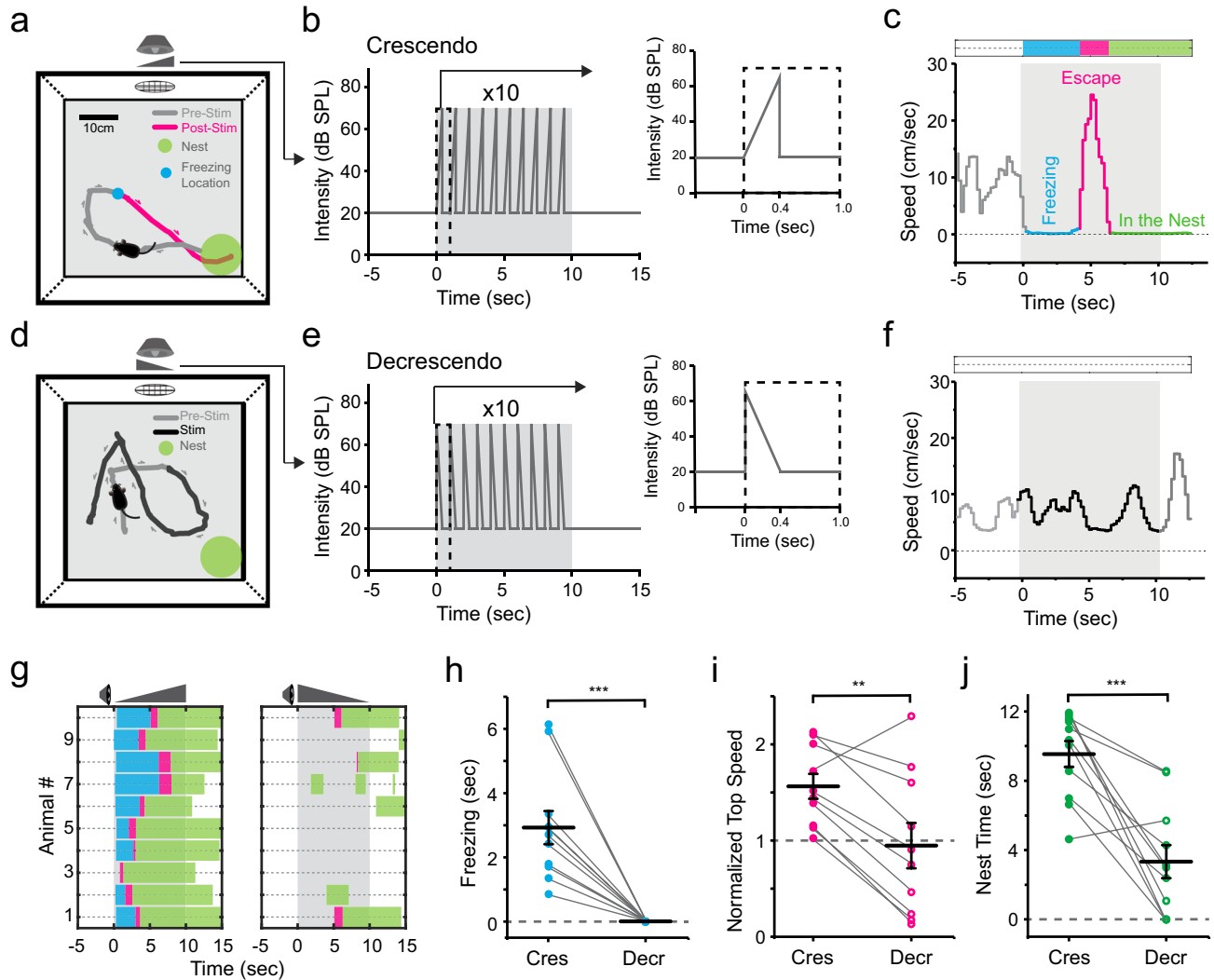

**Fig. 1 Looming sound-induced freezing-escape defensive behavior. a** Schematic diagram of open field test for sound-induced behavior. A speaker was attached to one of the walls. A nest (green circle) was placed in a corner. Gray and red traces label the movement trajectory before and after presenting (including during) crescendo stimuli respectively for an example animal. Blue dot marks the location of crescendo-triggered freezing. **b** Illustration of one trial of crescendo stimuli. Right: individual stimulus. **c** Locomotion speed for the example animal shown in (**a**). Different behavioral phases are labeled by different colors: freezing (blue), escape (red) and nest time (green). **d** Movement trajectory of an example animal in response to decrescendo stimuli. Gray: before decrescendo stimuli. Black: during presentation of decrescendo stimuli. **e** Illustration of decrescendo stimuli. **f** Locomotion speed for the animal shown in (**d**). **g**, Color-coded behavioral outcome in response to crescendo (left) and decrescendo (right) stimuli in 10 mice. **h** Comparison of crescendo and decrescendo induced freezing duration. Data points for the same animal are connected with a line. ***$p = 1.033 \times 10^{-4}$, two-sided paired $t$ test, $t = 5.669$, $n = 10$ mice. **i** Comparison of normalized top speed. **$p = 0.003$, two-sided paired $t$ test, $t = 4.059$, $n = 10$. **j** Comparison of nest time. ***$p = 6.103 \times 10^{-5}$, two-sided paired $t$ test, $t = 5.126$, $n = 10$. Data are presented as mean ± s.e.m. for (**h–j**). The center presents the mean value and the bar represents s.e.m.

duration (Fig. 1h, Decr), normalized top speed (Fig. 1i) and nest time (Fig. 1j). Decrescendo stimuli with a high-level background (70 dB SPL) also failed to produce a reliable defensive response (Supplementary Fig. 3). These results indicate that mice are much less fearful to decrescendo than crescendo stimuli. Therefore, the defense behaviors elicited are specific to sounds mimicking approaching objects.

**Looming-induced defense behaviors require activity of auditory cortex (AC).** Next, we explored potential neural circuits responsible for the crescendo-triggered freezing-escape behavior. As the AC has been implicated previously in a sound-induced flight behavior[17], we first investigated whether AC contributed to the crescendo-triggered freezing-escape behavior by silencing AC

with muscimol, an agonist of GABA$_A$ receptors (Fig. 2a). About 15–30 min after the injection of muscimol into AC, the open field test with crescendo stimuli was performed. Both the freezing duration and escape speed were significantly reduced compared to the control condition (Fig. 2b–c, see Supplementary Movie 2), while no effects were observed after injecting the control saline (Supplementary Fig. 4a). In particular, freezing was completely blocked by silencing AC in nearly all the mice tested (Fig. 2b–c). These results indicate that activity of AC is required for the crescendo-triggered defense behavior, including both freezing and escape.

We further examined spike responses of AC neurons to the crescendo sound stimulus, by performing single-cell loose-patch recording in AC in awake head-fixed mice, following our previous studies[22]. We focused on layer (L)5 since corticofugal neurons in

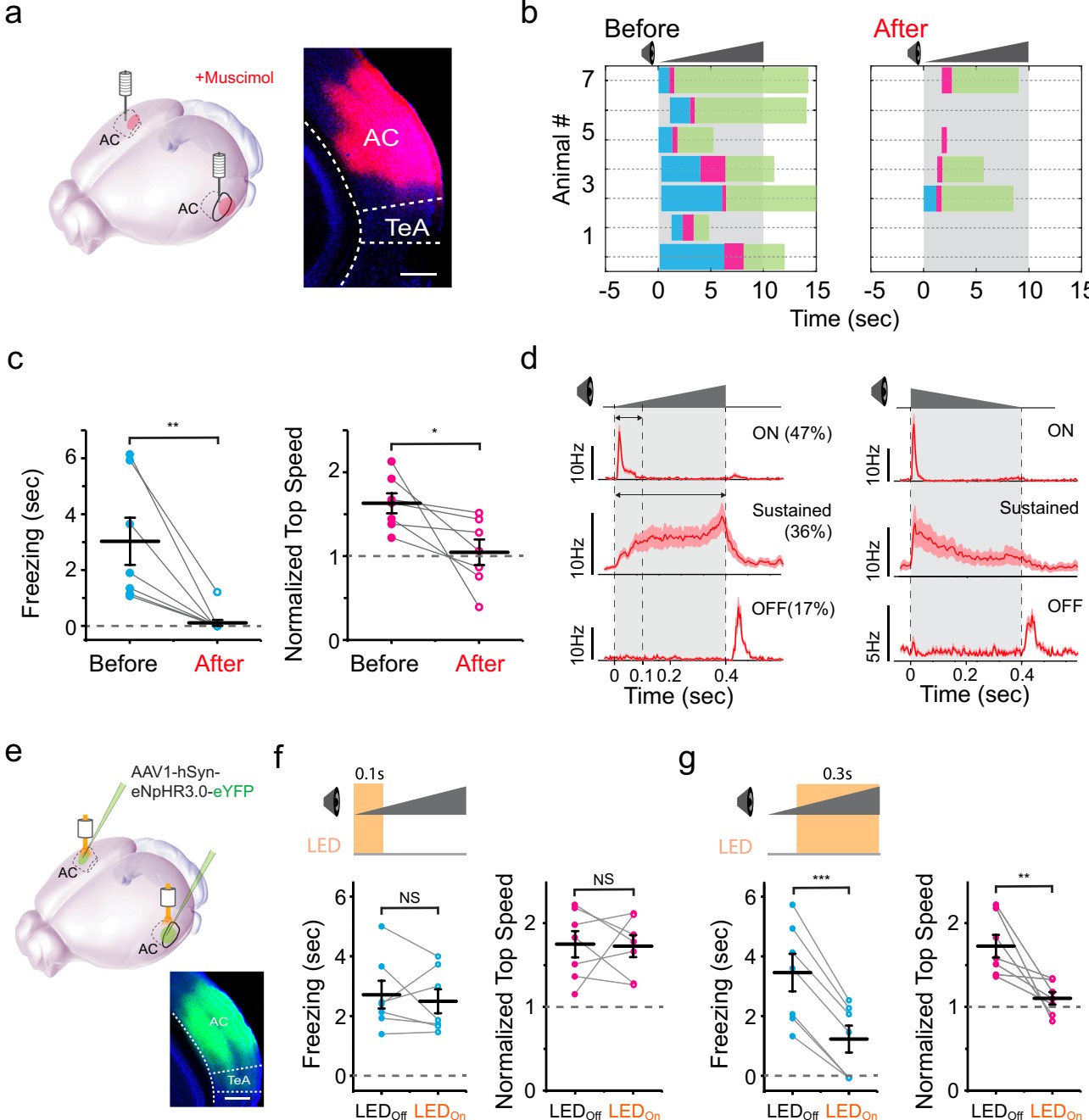

**Fig. 2 Crescendo-induced defensive behaviors require activity of auditory cortex (AC). a** Silencing AC with muscimol. Right, example image showing the spread of fluorescent muscimol in AC. Scale bar, 500 μm. TeA, temporal association cortex. **b** Crescendo-induced behavioral responses of seven mice two days before (left) and right after (right) silencing AC. **c** Comparison of crescendo-induced freezing duration (left, **$p = 0.010$, two-sided paired $t$ test, $t = 3.684$) and normalized top speed (right, *$p = 0.038$, two-sided paired $t$ test, $t = 2.652$) before and after silencing AC. Data points for the same animal are connected with a line ($n = 7$). Data are presented as mean ± s.e.m. The center presents the mean value and the bar represents s.e.m. **d** Population average of peri-stimulus spike time histograms (PSTHs) for AC layer 5 neurons exhibiting On, sustained and Off responses respectively ($n = 25, 21, 10$ cells) to the crescendo (left) and decrescendo (right) stimulus. **e**, Injection of AAV-hSyn-eNpHR3.0 in AC. Bottom, example image showing the expression of eNpHR3.0 in AC. Scale bar, 500 μm. **f**, Behavioral responses to crescendo stimuli without (LED$_{off}$) and with (LED$_{on}$) optogenetic inhibition of AC activity during 0-100 ms of each stimulus. Data points for the same animal are connected with a line ($n = 7$ mice). Left, $p = 0.59$, two-sided paired $t$ test, $t = 0.568$; right, $p = 0.90$, $t = 0.126$. N.S., non-significant. **g** Behavioral responses to crescendo stimuli without (LED$_{off}$) and with (LED$_{on}$) optogenetic inhibition of AC activity during 100-400 ms of each stimulus ($n = 7$ mice). Left, ***$p = 2.281 \times 10^{-5}$, two-sided paired $t$ test, $t = 10.42$; right, **$p = 0.007$, $t = 3.420$. Data are presented as mean ± s.e.m. for (**f-g**). The center presents the mean value and the bar represents s.e.m.

L5 give rise to subcortical projections to many targets associated with motor control[23]. Out of 47 recorded L5 neurons, 47% (22/47) exhibited transient On responses, 36% (17/47) exhibited sustained responses with gradually increasing firing rates, and 17% (8/47)

exhibited transient Off responses (Fig. 2d, left). These neurons displayed the same type of responses to the corresponding decrescendo stimulus, except that for the sustained type of cells the firing rate gradually decreased during sound (Fig. 2d, right).

The sustained and transient On responses exhibited similar onset latencies (sustained: $20.2 \pm 0.6$ ms; On: $21.6 \pm 0.4$ ms, $p = 0.68$, two-sided $t$ test). To address the question of whether these responses play differential roles in inducing defense behavior, we temporally silenced AC activity during different phases of the crescendo sound stimulus, by bilaterally injecting adeno-associated virus (AAV) encoding halorhodopsin (eNpHR3.0) in deep layers of AC and applying amber LED light (560 nm) (Fig. 2e). Optogenetic silencing during 0-100 ms of the sound, equivalent to silencing the transient On responses, had no effects on the crescendo-induced freezing or flight behavior (Fig. 2f). In contrast, silencing during 100-400 ms, i.e., silencing mainly the increasing sustained responses, largely reduced both the crescendo-induced freezing and escape (Fig. 2g). Based on these results, we conclude that the sustained responses of AC neurons, which exhibit differential activity temporal patterns to crescendo vs. decrescendo stimuli, play a critical role in driving crescendo-induced defense behaviors.

**Different roles of corticofugal targets**. We then wondered which structures downstream of AC were responsible for the crescendo-induced behaviors. Previously, SC has been shown to play a key role in both visual looming-induced freezing/flight[4–9] and loud sound-induced flight behavior[6,8]. Here, we silenced SC with muscimol (Fig. 3a) and found that the crescendo-induced escape was significantly impaired while freezing was not significantly affected (Fig. 3b-c, see Supplementary Movie 3 and saline control in Supplementary Fig. 4b). This result suggests that while SC plays a key role in mediating the escape behavior, another pathway downstream of AC is likely responsible for the crescendo-triggered freezing.

It appears that the neural pathway for the crescendo-triggered freezing is different from that underlying visually induced freezing, as SC is required for the latter but not for the former. We next examined potential targets downstream of AC, by injecting AAV encoding GFP into AC (Fig. 3d, left). Prominent GFP-labeled axons were observed in TS (Fig. 3d, right), consistent with previous reports of projections from AC to the striatum[24–26]. As passing axons through TS were noticed, to further confirm the AC-TS connectivity, we employed anterograde transsynaptic labeling[7] by injecting AAV1-Cre into AC of the Ai14 (Cre-dependent tdTomato) reporter mouse (Fig. 3e, left). TdTomato-labeled cell bodies were observed in the intermediate part of TS (Fig. 3e, middle and right), revealing an AC-recipient TS zone[25].

We next examined the functional role of the corticostriatal projection to TS by bilaterally injecting AAV-CaMKII-ChR2-YFP (or AAV-GFP as control) into the AC-recipient TS region (Fig. 3f). A train of blue light pulses (470 nm, 10-ms pulse duration, 20 Hz for 10 s) was delivered to TS during the open field test. The optogenetic activation of TS neurons alone significantly decreased locomotion of the mouse (Fig. 3g–h, see Supplementary Movie 4). In particular, the mouse directly froze during the first 5-s of photostimulation. No effect of LED light stimulation on locomotion was observed in the GFP control group (Supplementary Fig. 4c). These results suggest a potential involvement of TS in the freezing pathway.

**The corticostriatal projection to TS mediates looming-induced freezing**. To further test whether the corticostriatal pathway to TS is involved in the crescendo-induced freezing, we optogenetically inhibited TS by injecting AAV-hSyn-eNpHR3.0-eYFP bilaterally into TS and applying amber LED light (Fig. 4a). We found that with TS inhibited, the crescendo-triggered freezing nearly disappeared, and mice just rapidly escaped to the nest (Fig. 4b, see Supplementary Movie 5). While the freezing duration was

dramatically reduced, the escape speed was not affected but the escape latency was much shortened (Fig. 4c). These results indicate that TS activity is necessary for the crescendo-triggered freezing but not for the crescendo-triggered escape.

To further test whether the AC-TS pathway is sufficient for inducing freezing, we employed a two-step viral injection strategy[7,27] to specifically activate AC-recipient TS neurons. We first injected AAV1-Cre in AC, and then made a secondary injection of AAV encoding Cre-dependent ChR2 in TS (Fig. 4d). A train of blue light pulses (10-ms pulse duration at 20 Hz) was delivered to TS, covering the entire duration of the crescendo stimuli. The photoactivation of AC-recipient TS neurons dramatically prolonged the crescendo-triggered freezing and eliminated the escape phase (Fig. 4e–f, see GFP control in Supplementary Fig. 4d), and animals tended to freeze during the entire duration of the crescendo stimuli (Fig. 4e). Altogether, our behavioral results support the necessity and sufficiency of the AC-TS pathway activation for the defensive freezing.

**Differential temporal dynamics of TS and SC neuron responses**. Why does the looming-induced behavior exhibit a stereotyped sequence with freezing followed by flight? To address the question, we carried out multichannel recording (with single-unit sorting) with a silicon probe in SC or TS (Fig. 5a, see "Methods"). As shown by two example cells (Fig. 5b), SC neurons responded consistently to all of 10 crescendo stimuli, whereas TS neurons responded to the first few stimuli. The population average of evoked firing rates demonstrated that TS neuron responses rapidly declined with the increasing number of crescendo stimuli, whereas the responses of SC neurons showed much slower adaptation (Fig. 5c). On average, the responses of TS neurons to the first two stimuli were much stronger than those of SC neurons (evoked firing rate: $34.5 \pm 3.5$ Hz vs. $16.3 \pm 2.3$ Hz, ***$p < 0.001$, two-sided two-sample $t$ test, $t = 4.15$, $n = 55$, 46).

We reason that the initially strong but quickly adapted responses in TS neuron may account for the behavioral outcome that freezing is initially induced and then followed by escape behavior. Indeed, by comparing the change in evoked firing rate of TS neurons over time and the accumulative probability of escape latency (Fig. 5d), we found that the initiation of escape behavior was most likely to occur when TS neuron responses had rapidly declined (between 2 to 4 s). The decline of TC neuron responses was also temporally correlated with the decline of the probability of freezing (Fig. 5e). This supports the notion that TS neuron activity drives freezing. As mentioned above, cortical neurons with sustained but not transient responses are responsible for driving both the freezing and escape behaviors. Consistently, responses of these neurons as a population exhibited a time-dependent decline (Fig. 5f), although not as deep as TS neurons, whereas transient On/Off cortical responses remained more-or-less stable (Fig. 5g). This suggests that the adaption of TS neuron responses may be partially attributed to the adaptation of cortical neurons themselves.

To further understand the relationship between neuronal activity and behavioral outcome, we examined responses to looming sounds at different intensities. We found that changing the sound intensity from 50 to 90 dB SPL, there was an initial increase followed by a decrease in the evoked responses of TS neurons (Fig. 5h). This correlates with the initial increase and then decrease in the probability of freezing (Fig. 5i). In comparison, the evoked responses of SC neurons exhibited a consistent overall increase from 50 to 90 dB SPL (Fig. 5j), which correlates with the increase in the probability of escape (Fig. 5k). These results further confirm our conclusion that TS neuron activity underlies freezing and SC neuron activity underlies flight.

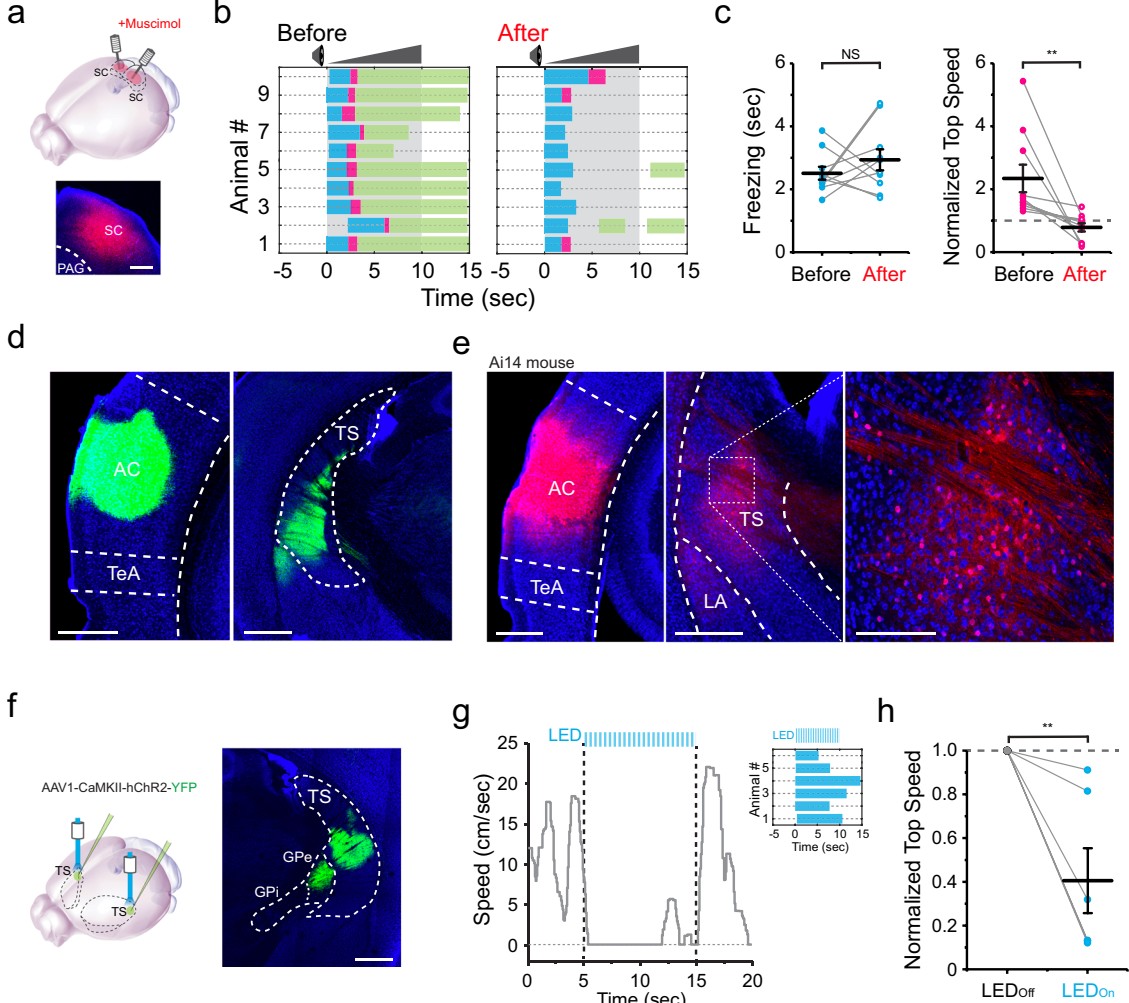

**Fig. 3 Differential roles of the superior colliculus (SC) and the tail of the striatum (TS) in the crescendo-induced defensive behavior. a** Silencing SC with muscimol. Scale bar, 200 μm. PAG, periaqueductal gray. **b** Crescendo-induced behavioral responses of 10 mice two days before (left) and right after (right) silencing SC. **c** Comparison of crescendo-induced freezing duration (left, $p = 0.358$, two-sided paired $t$ test, $t = -0.968$, $n = 10$ mice) and normalized top speed (right, $**p = 0.005$, two-sided paired $t$ test, $t = 3.239$, $n = 10$) before and after silencing SC. Data are presented as mean ± s.e.m. The center presents the mean value and the bar represents s.e.m. **d** Injection of AAV-GFP in AC. Left, expression at the injection site. Right, AC axons in TS. Scale bars, 500 μm. Blue, Nissl staining. AC auditory cortex; TeA temporal association cortex. **e** Injection of AAV1-hSyn-Cre in AC of Ai14 (Cre-dependent tdTomato) mice. Left, tdTomato (red) expression at the injection site. Middle and right (zoom in), transsynaptically labeled AC-recipient neurons in TS. Blue, Nissl staining. Scale bars, 500 (left and middle) and 100 (right) μm. LA, lateral amygdala. **f** Left, injection of AAV-CaMKII-ChR2 in TS. Right, ChR2 (green) expression in TS. Scale bar, 500 μm. GPe globus pallidus externa, GPi globus pallidus interna. **g** Plot of speed for an example animal in response to LED activation (10 s) of TS neurons. Inset, behavioral outcome for six mice. **h** Top speed (normalized to the top speed during 5-s window before stimulation trigger) in the LED off and LED on condition. $**p = 0.005$, two-sided paired $t$ test, $t = 4.002$. Data are presented as mean ± s.e.m. The center presents the mean value and the bar represents s.e.m.

**Corticofugal neurons targeting TS and SC are largely separate.** Since different corticofugal targets played different roles, we wondered whether the TS- and SC-projecting corticofugal neurons were separate neuronal populations. To address this question, we injected retrograde dyes (CTb) of different colors into TS and SC respectively in the same animal (Fig. 6a, b). The retrogradely labeled neurons in AC were predominantly observed in L5, with TC-projecting neurons across L5a and L5b while SC-projecting neurons mainly in L5b (Fig. 6c, left). We then quantified the percentage of cells in each group (red or green labeled) showing colocalization with the other color. We found that in both TS- and SC-projecting populations, the overlap was fairly small, at most a few percent (Fig. 6c, right). This indicates that TS- and SC-projecting corticofugal neurons are largely separate populations.

We also examined axon collaterals of TS-projecting cortical neurons, by first injecting AAV1-Cre[27] in TS and second injection of AAV encoding Cre-dependent GFP in AC (Fig. 6d). The retrogradely labeled AC neurons sent profuse axons to TS, while only very sparse axon collaterals were observed in SC (Fig. 6e). This result demonstrates that TS-projecting AC neurons do not have much influence on SC, further confirming the segregation of AC-TS and AC-SC pathways.

**Auditory corticofugal neurons preferentially target D2 MSNs in TS.** Finally, we noticed that AC axons projected to the intermediate part of the TS (Fig. 6e), where D2-type medium spiny neurons (MSNs) are reported to be clustered[28,29]. We thus wondered whether AC axons specifically innervated D2 MSNs

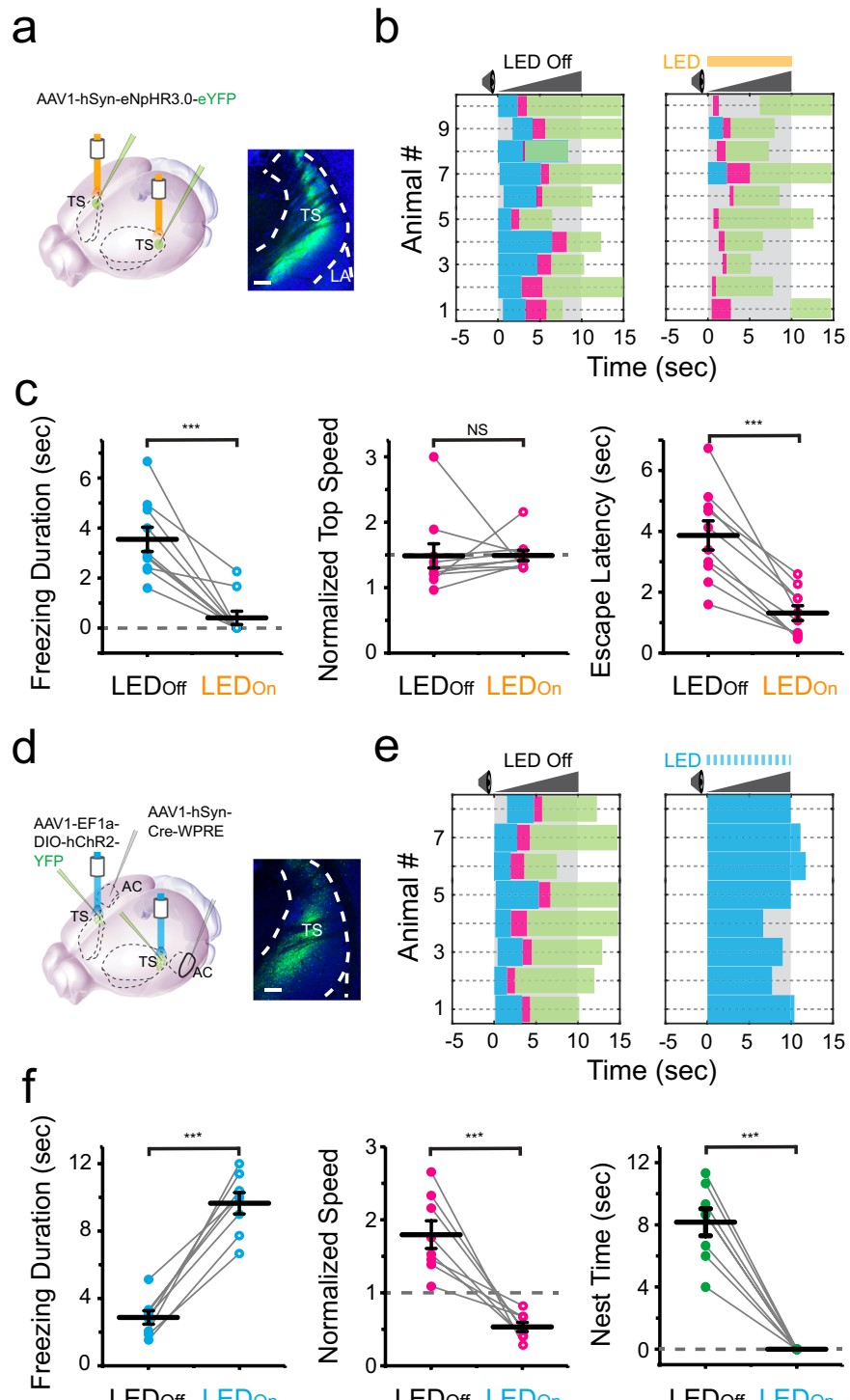

**Fig. 4 The corticostriatal projection to the tail of the striatum (TS) mediates crescendo-induced freezing. a** Left, injection of AAV-hSyn-eNpHR3.0 in TS. Right, eNpHR3.0 (green) expression in TS. Scale bar, 200 μm. LA lateral amygdala. **b** Behavioral responses to crescendo stimuli without (left) and with (right) optogenetic silencing of TS ($n = 10$ mice). **c** Comparison of crescendo-induced freezing duration (left, ***$p = 1.061 \times 10^{-4}$, two-sided paired $t$ test, $t = 5.962$, $n = 10$ mice), normalized top speed (middle, $p = 0.981$, two-sided paired $t$ test, $t = -0.024$) and escape latency (right, ***$p = 3.800 \times 10^{-5}$, two-sided paired $t$ test, $t = 6.835$) without (LED$_{off}$) and with (LED$_{on}$) silencing of TS. N.S., not significant. **d** Left, first injection of AAV1-Cre in AC and second injection of AAV1-DIO-ChR2 in TS. Right, expression of ChR2 in AC-recipient TS neurons. Scale bar, 200 μm. **e** Behavioral responses to crescendo stimuli without (left) and with (right) optogenetic activation of AC-recipient TS neurons ($n = 8$ mice). **f** Comparison of crescendo-induced freezing duration (left, ***$p = 0.006$, two-sided paired $t$ test, $t = -3.380$, $n = 8$ mice), normalized top speed (middle, ***$p = 0.001$, two-sided paired $t$ test, $t = 5.316$) and nest time (right, ***$p = 1.628 \times 10^{-5}$, two-sided paired $t$ test, $t = 9.381$) without (LED$_{off}$) and with (LED$_{on}$) coupling the activation of AC-recipient TS neurons.

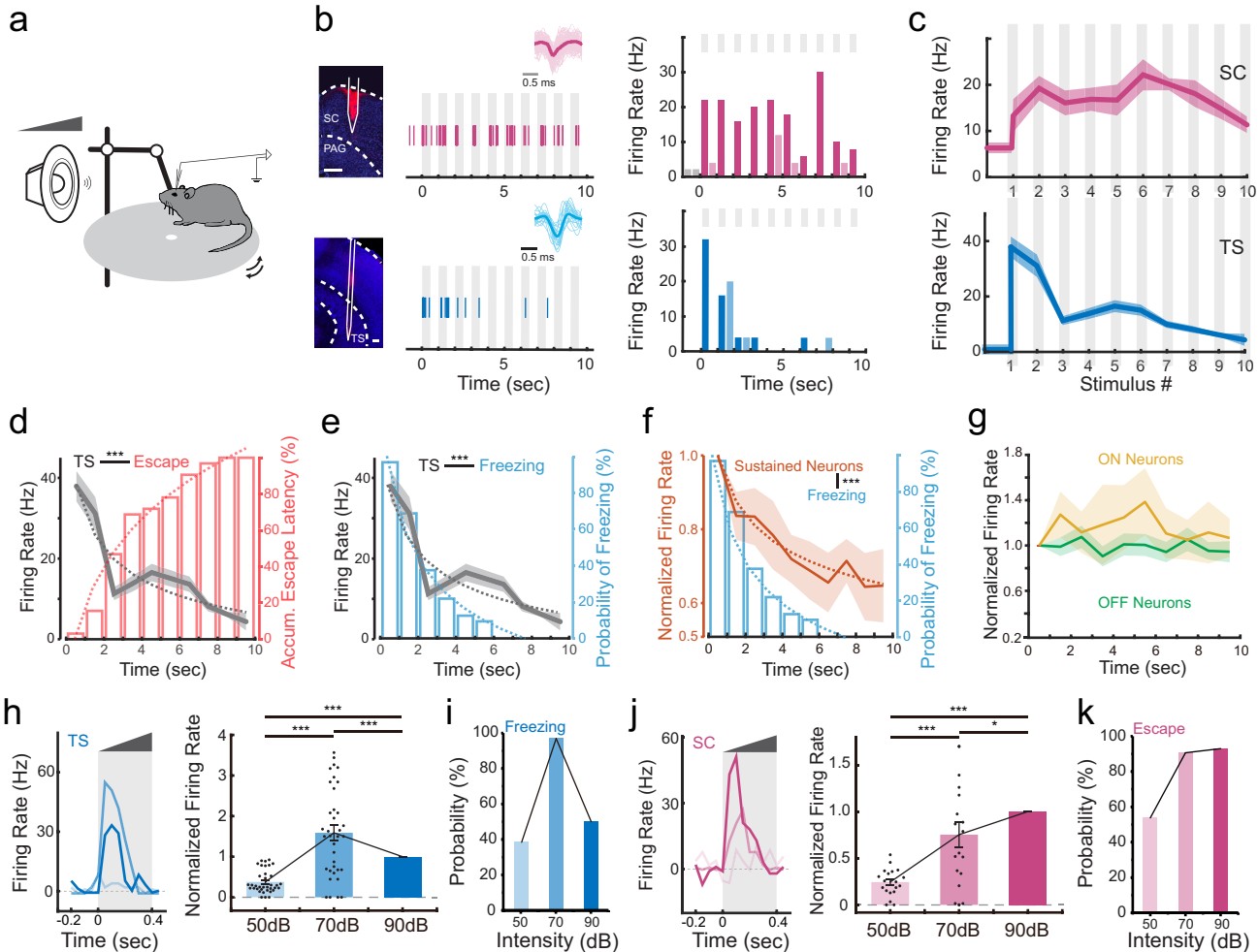

**Fig. 5 Differential temporal dynamics of superior colliculus (SC) and the tail of the striatum (TS) neuron responses to crescendo stimuli. a** Illustration of in vivo recording in the awake head-fixed condition. **b** Left, example image showing track of the recording electrode in SC (top) or TS (bottom). Scale bars, 200 μm. Middle, example single-unit response to a train of crescendo stimuli in SC (top) or TS (bottom). Each vertical line represents one spike. Grey column labels the duration of a sound stimulus. Superimposed individual spike waveforms and the average are shown in top insets (SC, $n = 104$ spikes; TS, $n = 22$ spikes). Right, corresponding PSTH. Dark and light-shaded colors represent spike rates during a sound stimulus (marked by grey column) and inter-stimulus intervals respectively. PAG periaqueductal gray. **c** Population average of evoked firing rate (analyzed for the duration of each individual stimulus) across10 crescendo stimuli ($n = 46$ and 55 for SC and TS neurons respectively). Shade indicates s.e.m. **d** Comparison of the mean evoked firing rate of TS neurons across 10 stimuli and accumulative distribution of escape latencies. Dotted curve represents fitting of the data. ***$p < 0.001$, $r = -0.92$, Pearson's correlation coefficient. **e** Comparison of the mean evoked firing rate of TS neurons across 10 stimuli and the probability of freezing over time. ***$p < 0.001$, $r = 0.92$, Pearson's correlation coefficient. **f** Comparison of the mean evoked firing of the sustained type of cortical L5 neurons and the probability of freezing over time. ***$p < 0.001$, $r = 0.95$, Pearson's correlation coefficient. **g** Comparison of the mean evoked firing of the ON (yellow) and OFF (green) type of cortical neurons across 10 stimuli. ON neuron response vs. probability of freezing, $p = 0.58$, $r = 0.20$; OFF neuron response vs. probability of freezing, $p = 0.74$, $r = -0.12$, Pearson's correlation coefficient. **h**, Evoked firing rate of an example TS neuron (left) and normalized responses of all the recorded TS neurons (right) to crescendo stimuli with different top intensities. Right, ***$p < 0.001$, one-way ANOVA with post hoc (Bonferroni) test, $n = 36$. **i** The probability of freezing to crescendo stimuli with different top intensities ($n = 13$, 32 and 14 mice respectively). **j** Evoked firing rate of an example SC neuron (left) and normalized responses of all the recorded SC neurons (right) to crescendo stimuli with different top intensities. Right, ***$p < 0.001$, *$p = 0.022$, one-way ANOVA with post hoc (Bonferroni) test, $n = 19$. **k** The probability of escape to crescendo stimuli with different top intensities ($n = 13$, 32 and 14 mice respectively).

in TS. To address this question, we transsynaptically labeled AC-recipient TS neurons by injecting AAV1-Cre in AC of Ai14 mice (see Fig. 3e). In brain slices, we then performed RNAscope assay[30] using probes for detecting expression of D1 and D2 dopamine receptors respectively (see "Methods"). We found that in TS the D1+ and D2+ positive cells were largely spatially segregated (Fig. 6f). The tdTomato-labeled AC-recipient TS neurons were mainly located in the D2-enriched domain (Fig. 6g). In this cell population, about 70% expressed D2 receptors while only about 20% expressed D1 receptors (Fig. 6h).

These results demonstrate that auditory corticofugal axons projecting to TS preferentially target the D2 type of MSNs.

## Discussion
In this study, we have demonstrated that auditory looming stimuli, similar to visual looming stimuli, can invoke defense-like behaviors in freely moving mice. These behaviors are characterized by a stereotypical freezing followed by escape sequence, which are mediated by different corticofugal targets, TS and SC,

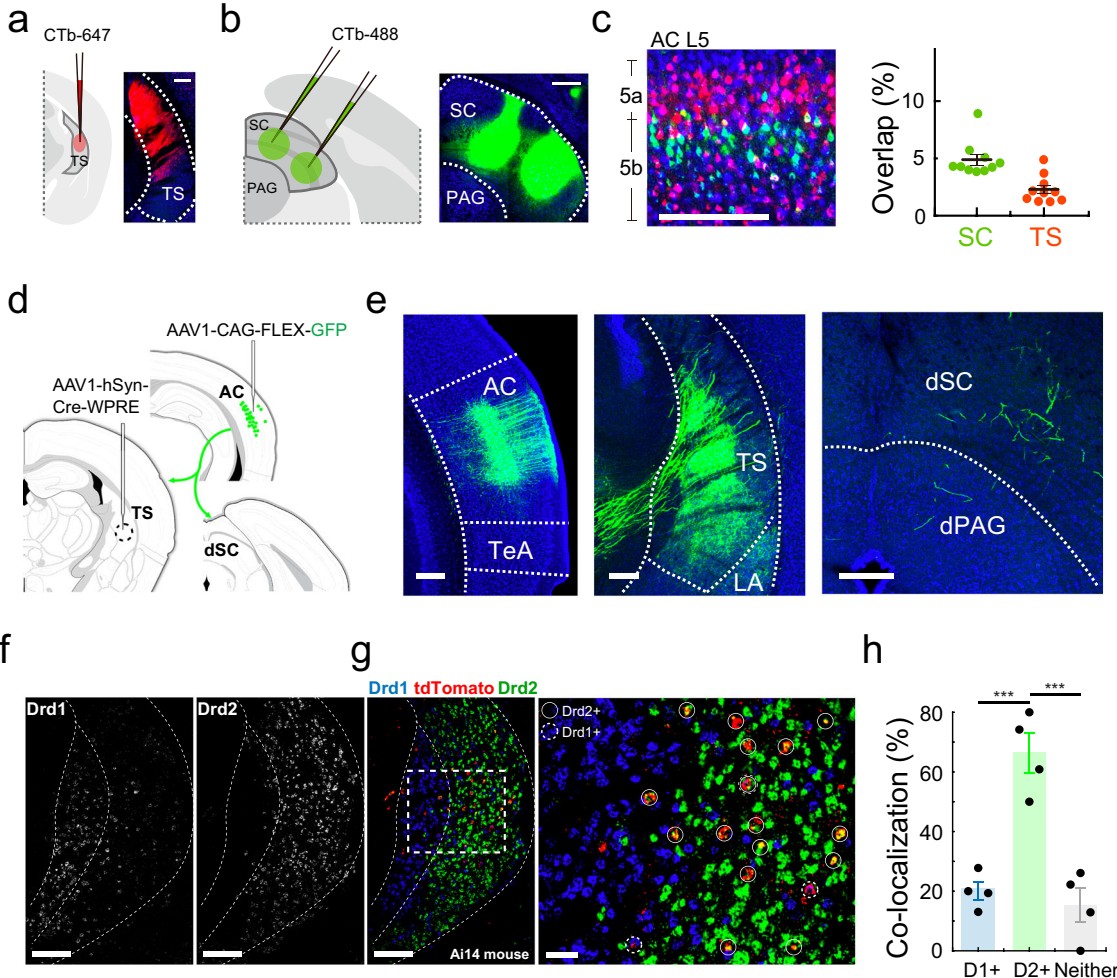

**Fig. 6 SC- and TS- projecting corticofugal neurons and targets. a** Injection of CTb-647 (red) into TS. Scale, 200 μm. TS the tail of the striatum. **b** Injection of CTb-488 (green) in SC in the same animal. Scale, 200 μm. SC superior colliculus, PAG periaqueductal gray. **c** Left, retrogradely labeled TS-projecting (red) and SC-projecting (green) neurons in layer 5 (L5) of auditory cortex (AC). Scale bars, 200 μm. Right, overlap between the two groups of projecting neurons (overlap ratio: 4.9 ± 0.5% in SC-projecting neurons. 2.3 ± 0.4% in TS-projecting neurons, $n = 10$ sections). Data are presented as mean ± s.e.m. The center presents the mean value and the bar represents s.e.m. **d** Viral injections in TS and AC to label TS-projecting AC neurons. **e** Retrogradely labeled TS-projecting neurons in AC (left) and their axon terminals in TS (middle) and SC (right). Scale bars, 200 μm. TeA temporal association cortex, LA lateral amygdala, dSC deep layers of the superior colliculus, dPAG dorsal periaqueductal gray. The result was consistent in three mice. **f** Distribution of Drd1+ (left) and Drd2+ (right) neurons in TS. Scale bars, 200 μm. **g** AC-receipt neurons (tdTomato, red) and Drd1+ (blue) or Drd2+ (green) neurons in TS. Scale bars, 200 (left) and 50 (right) μm. **h** Percentage of AC-receipt neurons overlapping with D1+(i.e. Drd1+) neurons, D2+ (i.e. Drd2+) neurons or neither. D1+vs. D2+, ***$p = 5.875 \times 10^{-4}$; D2+vs. Neither, ***$p = 2.827 \times 10^{-4}$, one-way ANOVA with post hoc (Bonferroni) test, $n = 4$ sections. Data are presented as mean ± s.e.m. The center presents the mean value and the bar represents s.e.m.

respectively. The transition from freezing to escape is temporally correlated with a rapid adaptation of TS neuron responses to repeated presentations of looming stimuli. Our study establishes a strong correlation between activity patterns of corticofugal circuits and defense behavior outcomes.

**Crescendo sound for inducing defense behaviors**. The crescendo/looming sound we used was broadband white noise generated on top of a noise background. The white noise is commonly used as a general-purpose testing sound. It is very different from the communication sounds (either short-range or long-range) of mice[31] in terms of frequency range and spectrotemporal structures. For example, mouse communication sounds consist of ultrasonic frequency-modulated (FM) sweeps[31–33]. Defense-like behaviors triggered by crescendo sounds are most likely responses to the danger signals cued by the temporal change of sound intensity rather than potential physical harms

due to a loud sound, since decrescendo stimuli of the same peak intensity failed to induce either freezing or escape. In addition, we examined behavioral adaptation under loud crescendo sounds (90–100 dB SPL). The probability of freezing dropped from 56% (five out of nine animals) in the first trial to 33% (3/9) in the second trial, and the probability of escape dropped from 100% (9/9) to 44% (4/9). Such behavioral adaptation can be well explained by the habituation of animals to auditory cues of threats but is difficult to be explained by responses to potential physical harms caused by the sound.

Following previous visual looming studies[4–10], we have applied repeated presentations of ten looming stimuli, which may not be an often-encountered situation in natural environments. In a separate set of experiments, we have applied only one crescendo stimulus (70 dB SPL) with a long rise time (5 s). This single stimulus was effective in inducing defense behavior (either freezing or escape) in 16 out of 21 animals (freezing duration: 3.2 ± 0.5 s; normalized top speed: 1.6 ± 0.2) with freezing-escape

sequence observed in four mice. Nevertheless, it appears that the repeated pattern of stimulation is more effective and robust ($p = 0.0071$, Fisher's exact test) for inducing defense behavior, which could be due to its higher effectiveness in attracting the animal's attention.

Previous work has shown that simple auditory stimuli can elicit fearful responses even in the absence of the AC, due to the connections of the auditory thalamus to other areas, such as the amygdala[34]. Defense behaviors invoked by more complex stimuli, however, appear to require the AC[14,34]. Here, we show that AC is required for both crescendo-induced freezing and escape. It is possible that the relatively high complexity of the crescendo type of auditory stimuli requires the AC to encode the threat signals.

**Functional properties of L5 corticofugal neurons**. Corticofugal neurons in L5 project to many subcortical targets associated with motor control, and the L5 output represents the only known substrate by which the cortex can directly influence behavior[23]. In the present study, we show that corticofugal neurons projecting to TS and SC are largely separate populations. Although the two corticofugal pathways are separate, with one controlling freezing and the other controlling flight, they may have competitive interactions at some lower-level nuclei which remain to be identified. According to electrophysiological properties, L5 pyramidal neurons can be categorized into "regular spiking" (RS) and "intrinsic bursting" (IB) cells[35–37]. Previously, in the rat primary AC, we have reported that IB neurons in general have broader frequency tuning than RS neurons[38]. Here, following previous studies[38–42], we have categorized the recorded L5 neurons into RS and IB subpopulations (Supplementary Fig. 5a, b). Consistent with previous findings in the rat[38], we found that the IB subpopulation exhibited broader frequency-intensity tonal receptive fields, higher spontaneous and evoked firing rates, and slightly faster tone responses than the RS subpopulation (Supplementary Fig. 5c–e). In addition, we found that a majority of IB neurons (55%) exhibited the sustained response type while few of RS neurons (10%) exhibited this response type. Since cortical neurons with sustained responses mediate the induced defense behaviors, we conclude that out of L5 corticofugal neurons the IB neurons contribute the most to the auditory looming-induced defense.

**SC mediates looming sound-induced flight**. For visual looming-induced defense, SC is required for both freezing and flight behaviors[10], whereas for auditory looming-induced defense, SC is not required for freezing. This difference may be due to different positions of SC in visual vs. auditory defense circuits. SC is the gateway for the majority of incoming visual information, but only receives partial auditory information from AC and other areas. Therefore, SC takes a more downstream position for auditory than visual defense responses. SC has been shown to play an important role in defensive flight behavior by a number of previous studies[4–9], likely by activating the PAG[6]. Here, our result showing that SC is responsible for auditory looming-induced escape behavior is consistent with these previous studies. It is also consistent with our previous report that activating AC-recipient SC neurons directly causes flight behavior[7].

**TS mediates looming sound-induced freezing**. Different from visual looming results, here we find that freezing induced by auditory looming stimuli is driven by the corticostriatal projection to TS. Consistent with previous findings[24,25,28], we show in the current study that there is a spatially specific projection from AC to the intermediate part of the TS, where D2-type MSNs are clustered. The AC axons also preferentially innervate D2-type MSNs.

Consistently, the AC-recipient TS neurons project to the external globus pallidus (GPe) (Fig. 3f and Fig. 6g), which is known to be part of the D2/indirect pathway[27,43–47]. Previously, it has been shown that activation of D2-type MSNs directly suppresses locomotion of the mouse[48]. Therefore, the revealed AC-TS pathway can drive D2-type MSNs in TS, which can account for the TS-dependent crescendo-induced freezing.

It is likely that the corticofugal pathways through TS and SC compete at some level of downstream structures directly controlling motor functions, e.g., the PAG. We show that TS and SC neuron responses display different dynamics during repeated exposures to looming stimuli. The TS-mediated pathway initially has an advantage since TS neuron firing rates are much higher than SC neurons. However, TS neuron responses become quickly adapted over time, while SC neuron responses are somewhat facilitatory or remain at a relatively constant level. As TS neuron responses are greatly reduced, SC neuron responses become more dominant, leading to a transition from the freezing to escape phase of the behavior.

TS, which is also termed the caudal extreme domain of the caudoputamen (CPc.exe)[25], appears to be an anatomically and functionally unique region of the striatum[49,50]. Not only it is densely innervated by projections from the primary AC[25,29,51], but also it receives dopaminergic axons which are found to respond to aversive stimuli[49,52] and therefore may play a role in the reinforcement of avoidance[49]. While previous auditory studies on TS have been focused on its involvement in reward-motivated auditory discrimination tasks and acquisition of sound-action associations[26,53,54], our present study provides a new view on TS' role in delivering innately threatening auditory signals and driving auditory-induced defensive behavior.

## Methods
All experimental procedures in this study have been approved by the Institutional Animal Care and Use Committee (IACUC) of the University of Southern California.

**Animals**. Mice were housed at 18–23 °C with 40–60% humidity in a 12-h light-dark cycle (6AM-6PM) with ad libitum access to food and water. Male and female adult (2-3 months age) C57BL/6J mice and Ai14 (Cre-dependent tdTomato reporter; Jackson Laboratories, RRID: MSR_JAX:007914) were used in this study.

**Implantation of drug cannulas**. At least one week before the behavioral test, mice were prepared for cannula implantation. The animal was anaesthetized with iso-flurane (1.5% by volume in oxygen) and fixed on a stereotaxic frame (Kopf Instrument). After a craniotomy, two cannulas (OD = 0.48 mm, RWD) were implanted bilaterally for injection into AC (3.0 mm posterior and 4.5 mm lateral to bregma and 0.75 mm ventral from the cortical surface) or SC (4.0 mm posterior and 0.8 mm lateral to bregma and 1.5 mm ventral from the cortical surface). Coordinates were based on the Allen Reference Atlas. Implants were affixed by dental cement. Ketoprofen (0.5 mg/kg) was injected subcutaneously before they were returned to home cages and also in the next two days. After all the behavioral tests, mice were euthanized to verify the location of drug cannulas.

**Virus injection and optical fiber implantation**. For anterograde terminal and transneuronal labeling, AAV1 encoding CB7-CI-eGFP-WPRE (60 nL, UPenn Vector Core, $4.2 \times 10^{13}$ GC/mL) or hSyn-Cre-WPRE (80 nL, UPenn Vector Core, $2.5 \times 10^{13}$ GC/mL) was injected into AC of C57BL/6J or Ai14 mice. For TS silencing or activation, AAV1-hSyn-eNpHR3.0-eYFP (Addgene, $2 \times 10^{13}$ GC/mL) or AAV1-CaMKII-hChR2-YFP (UPenn Vector Core, $2.5 \times 10^{13}$ GC/mL) was injected into TS of C57BL/6 J mice (50 nL) bilaterally. For transneuronal activation, AAV1-hSyn-Cre-WPRE was injected bilaterally into AC (80 nL). Following 2–3 days, a second injection of AAV1-EF1a-DIO-hChR2-YFP (UPenn Vector Core, $1.6 \times 10^{13}$ GC/mL) was made in TS (60 nL). Ketoprofen (0.5 mg/kg) was injected subcutaneously into the mice before they were returned to home cages and in the next two days.

Optical fibers were implanted into TS two weeks after the virus injection. Surgery procedure was similar to drug cannula implantation. Optical fibers (NA = 0.22, RWD) were implanted bilaterally into the TS (1.0 mm posterior and 3.1 mm lateral to bregma and 2.6 mm ventral from the cortical surface) based on the Allen Reference Atlas. The mice were euthanized to verify the location of viral expression and optical fibers after all the behavioral tests.

**Activity manipulation.** In this study, we applied a similar method to silence AC or SC as previously described[17]. For pharmacological silencing, fluorescence-conjugated muscimol (1.5 mM, Life Technologies) was used. The pipette for injection was inserted through the implanted drug cannula, and muscimol (about 100–150 nL) was slowly (~3 min) injected into the targeted region. The behaviors were tested at 15–30 min after muscimol injection. After the behavioral tests, the mice were euthanized to verify the spread of muscimol. For control, we injected 100 nL saline into the AC or SC and performed the behavioral test 15 min later.

For optogenetic silencing or activation, optic fibers (NA = 0.22, 200-μm diameter, Thorlabs) connecting to an amber or blue LED light source (560 and 470 nm, respectively, Thorlabs) were connected to the implanted fibers in the targeted regions. The light power was set to about 7–10 mW (measured from the fiber tip) for cell body stimulation. For optogenetic silencing of TS, amber light was delivered into the implanted fibers simultaneously with the auditory stimuli. For optogenetic activation of TS without auditory stimuli, a blue light train (10-ms pulse duration at 20 Hz) was delivered into the implanted fibers during an open field test. The blue light train was turned ON for 0.4 s and OFF in the following 0.6 s to simulate crescendo stimuli. This light activation was repeated 10 times (10 s in total) in one behavioral trial. For optogenetic silencing of cortical neurons, amber light of 0.1 or 0.3 s was coupled with each auditory stimulus. For optogenetic activation of TS (without or with coupling with sound), blue LED pulses (25-ms pulse duration, at 20 Hz) were applied for 0.4 s and then off for 0.6 s, which were repeated for 10 times. For optogenetic silencing of TS, continuous amber light was delivered into the implanted fibers during the entire duration of auditory stimuli (including inter-stimulus intervals). For control, we applied the same LED light simulation in mice injected with AAV-GFP and performed the same behavioral tests.

**Auditory stimuli.** For crescendo stimuli (broadband white noise), noise intensity was increased linearly from 20 to 70 dB SPL within 0.4 s and then remained at 20 dB SPL for 0.6 s, and this was repeated for 10 times (Fig. 1b). Other peak intensities (50, 70, 90, and 100 dB SPL) and rise times (0, 0.2, 0.4, 0.8, and 4.0 s) were also tested (Supplementary Figs. 1–2). The speaker (XT25G30-04 1" Dual Ring Tweeter; Vifa) calibrated for a frequency range of 1–60 kHz was used to deliver auditory stimuli. The auditory stimulus was repeated 10 times except for the 4.0-s rise time experiment where there were two repeats. In a separate experiment, one crescendo stimulus with 5.0-s rise time was applied. One trial of auditory stimuli was presented in one day for each mouse. To test adaptation, five trials were presented in one day with 2–5 min intervals. For decrescendo stimuli with 20 dB SPL noise background, the intensity of noise was increased linearly from 20 to 70 dB SPL within 5 ms and then decreased linearly from 70 to 20 dB SPL in 0.4 s and remained at 20 dB SPL in the following 0.6 s, in a block (Fig. 1e). For decrescendo stimuli with 70 dB SPL noise background, a 70-dB noise was presented during habituation as background. When decrescendo was triggered, the noise intensity was decreased linearly from 70 to 20 dB SPL in 0.4 s and then retuned to 70 dB SPL within 5 ms. The intensity was kept at 70 dB SPL during the subsequent 0.6-s interval as well as after the stimuli (Supplementary Fig. 3a).

**Behavioral tests.** Behavioral tests were all performed in a sound-attenuation booth. An open-top box (in cm: 48 w, 48 l, 47 h) was used for the open field test. There was a hole on each of the four opaque walls, which was covered by mesh. A speaker was randomly attached to one of the holes for delivering sound. The noise sound was generated with LabVIEW (PCI-6040, 12-bits output, 1 MHz sampling rate, National Instruments) and measured as previously described[17]. A shelter nest was placed in the box at one corner and the crescendo/decrescendo stimuli were triggered manually. One camera was mounted above the box to monitor the locomotion of the mouse.

The animal was placed in the box and given 8–10 min to explore the environment. After the habituation period, auditory stimuli were triggered when the mouse entered a square zone in the middle of the arena. When two types of stimuli (crescendo vs. decrescendo) or two conditions (LED ON vs. LED OFF in optogenetic manipulation) were tested in the same mouse, we randomized the test sequence with one performed at least 48 h after the other. In the experiments of silencing AC or SC with muscimol, we tested the normal condition first and then infused AC or SC with muscimol or saline 48 h later.

**Behavioral quantification and data analysis.** The mouse behavior was recorded by the camera mounted above the testing box during the habituation period and auditory presentation. Locations and moving speeds of mice were calculated based on the recorded videos at 20 or 14 frame/s and analyzed offline by customized MATLAB (Mathworks) scripts with Tracker, a free and open-source video analysis and modeling software.

Freezing and escape were evaluated based on the behavioral videos and locomotion speed of the mouse. Averaged and top locomotion speeds of the mouse during the habituation were calculated as baseline. Freezing was defined as an episode of 1 s or more when the transient locomotion speed was consistently below 2 cm/s. Escape was defined as an episode of mouse running into the corner or the nest without stopping. Moreover, to distinguish the defensive escape from random exploration, the running speed had to be 10% higher than the top speed during the 5 s before the presentation of auditory stimuli.

To quantitatively analyze the duration of freezing and latency of escape, the onset timings of freezing and escape were determined based on the behavioral video frames. The onset of freezing was defined as the time when the movement of the mouse stopped completely after starting presenting auditory stimuli, and that of escape was determined as the first video frame when the mouse head turned followed by a full-body turn and running towards the nest.

To quantify the change in locomotion evoked by auditory stimuli or optogenetic manipulation, we measured a normalized top speed, which was defined by the peak speed during the presentation of stimuli normalized to that during the 5-s window before the stimuli. Nest time was quantified as the time the mouse spent in the shelter or corner during the 15-s window after the presentation of auditory stimuli.

**In vivo awake recording and data analysis.** Under the anesthesia with isoflurane (1.5% by volume in oxygen), a metal post for head fixation during the following recording experiment was mounted onto the skull of the animal and fixed with dental cement. The animal was trained to run freely on the running plate of the recording setup in the head-fixed condition. A craniotomy was performed over recording regions (AC, SC or TS) after anesthesia, and after the surgery a silicon elastomer (Kwik-CAST, WPI) was applied to cover the surgical opening before recording experiments.

Recordings were performed in a sound-attenuation booth (Acoustic Systems). For each animal, 1–2 recording sessions were applied every day with each session lasting for no more than 1 h. The same auditory stimuli that we used in the open field test were represented in the recording sessions. Before the recording, the silicon seal was removed. A 64-channel silicone probe (NeuroNexus, coated with diI) was used to penetrate into the desired brain region. We used an Open-Ephys system (Open Ephys) to record signals at 30 kHz sampling rate and saved the raw data for offline spike sorting and analysis. To obtain single-unit spikes, a semiautomatic spike sorting was performed offline by using Offline Sorter (Plexon), following our previous studies[55,56]. Single-unit activities were analyzed with customized MATLAB scripts. After all recording sessions, mice were euthanized to verify the location of the electrode.

For single-cell loose-patch recording from AC, patch pipettes (Kimax) with 5~7 MΩ impedance filled with artificial cerebrospinal fluid (ACSF: 124 mM NaCl, 1.2 mM NaH$_2$PO$_4$, 2.5 mM KCl, 25 mM NaHCO$_3$, 20 mM glucose, 2 mM CaCl$_2$, 1 mM MgCl$_2$) were inserted into AC. Agar (3.5% in warm ACSF) was put on to minimize cortical pulsation. Recordings were made with an Axopatch 200B amplifier (Molecular Devices). A loose seal (100–500 MΩ) was formed on the cell body, allowing spikes only from the patched cell to be recorded. Spike responses were recorded under the voltage-clamp mode, with the command potential adjusted so that a near 0-pA baseline current was achieved. Signals were sampled at 20 kHz. Blind patch recording with relatively large pipette opening sizes used here has a strong sampling bias towards pyramidal neurons. We recorded neurons located at 525–700 μm below the pia, corresponding to layer 5, in the primary AC based on tonotopic gradients mapped beforehand. The cells of which the spontaneous spike was zero within 10 min would not be further recorded. Mapping of tonal receptive field (TRF) was performed similarly to what we described previously[17,22,57–61], by presenting pure tones (100 ms) of various frequencies (2–32 kHz, 0.2-octave steps) and intensities (10–70 dB SPL, 10 dB steps). RS and IB neurons were categorized based on the responses to best-frequency tones: RS neurons exhibited only one spike while IB neurons exhibited a train of multiple spikes to the tone stimulation[38]. TRF was plotted as a heatmap using custom MATLAB scripts. Bandwidth of TRF was measured at 20 dB above the intensity threshold. Evoked firing rate was calculated for responses to best-frequency tones within a 100-ms window after the onset of tones. Spontaneous firing rate was calculated within a 50 ms before tone onsets. Response onset latency was determined from the PSTH at the time point when the firing rate exceeded the baseline firing by 3 standard deviations.

**Histology and imaging.** To verify the spread of muscimol, viral expression and electrode location, animals were deeply anesthetized with isoflurane and trans-cardially perfused with 4% paraformaldehyde (PFA) in phosphate-buffered saline (PBS). In the anatomical experiments, the mice were euthanized three weeks following injection. The brain tissue was sliced into coronal sections (150-μm thickness) by using a vibratome (Leica Microsystems). The sections were then stained with Nissl reagent (Deep red, Invitrogen) for 4 h at room temperature. Slices were imaged using a confocal microscope (Olympus FluoView FV1000). A ×4 objective was used to verify the muscimol spread or viral expression. Axonal labeling or transsynaptic labeling in areas of interest was further imaged under a ×10 objective.

**RNAscope assay.** We used the RNAscope Multiplex Fluorescent Reagent Kit v2 (ACD) and followed its user manual closely (Document Number 323100-USM, Rev Date: 02272019). We used probes Mm-Drd1a, Mm-Drd2-C2 and M-tdTomato-C3 to target Drd1, Drd2 and tdTomato mRNA respectively, and 1:1500 dilution for the Opal fluorophores.

Ai14 mice were sacrificed 3 weeks after injecting AAV1-hSyn-Cre-WPRE in AC (3.0 mm posterior and 4.5 mm lateral to bregma and 0.75 mm ventral from the cortical surface). The brain tissue was fixed in 4% PFA for 24 h at 4 °C. After

dehydration in 10%, 20% and 30% sucrose for 24 h successively, the tissue was sliced into 30-μm-thick coronal sections using a cryostat (CM3050S, Leica). The brain sections were mounted onto pre-cleaned slides (VWR Micro Slides, Superfrost Plus, Cat No. 48311-703). After washing the optimal cutting temperature (OCT) compound away with PBS, sections were baked for 30 min at 60 °C and post-fixed in 4% PFA for 15 min at 4 °C. Then, sections were subsequently dehydrated in 50%, 70%, and 100% EtOH, followed by incubation of hydrogen peroxide for 10 min at room temperature (RT). Next, target retrieval was performed by immersing the slides into distilled water for 10 s and into Retrieval Reagent for 5 min at 99 °C in a humidity saturated environment. The slides were cooled in distilled water for 15 s, transferred to 100% alcohol for 3 min at RT and fully dried. The following incubations were all done using the HybEZ Humidifying System (ACD). Between hybridizations, sections were washed in Wash Buffer for 2 min at RT. Next, the sections were incubated in Protease III for 30 min at 40 °C, followed by wash with distilled water. Probes mix was hybridized for 2 h at 40 °C. Then, amplification probes (AMP1, 2, and 3) were hybridized at 40 °C (AMP 1 and 2 for 30 min; AMP 3 for 15 min) sequentially. Finally, HRP probes (HRP-C1, HRP-C2 or HRP-C3) were hybridized for 15 min at 40 °C, followed by 30 min incubation with Opal fluorescent ligands (Opal 520, Opal 690 or Opal 570, 1:1500 dilution) for 30 min at 40 °C sequentially. The HRP blocker was applied for 15 min at 40 °C between each HRP probe hybridization. Sections were eventually mounted with DAPI and 60% glycerol in PBS and imaged under a ×10 objective using confocal microscope.

**Statistics**. In the behavioral experiments, sample sizes were selected based on previous related experiments and the literature and verified by power analysis. A power analysis was also used to determine sample sizes in the electrophysiological recordings. The sample sizes of anatomical and RNAscope results were the maximal available of our dataset. All related data are included in analysis. There is no exclusion of data in this study. Animals were randomly assigned to control and treatment groups. For the animals with multiple treatment, the sequence of treatment was randomized. In this study, the investigators were not blinded to group allocation during behavioral and electrophysiological data collection, since the investigators must adjust and trigger different auditory stimuli. Data analyses were performed blinding to experimental conditions. For anatomical experiments, the investigators were not blinded to group allocation during data collection and/or analysis as no grouping in this study. Paired $t$ test was used to compare data from the same animal. One-way ANOVA test and post hoc Bonferroni's multiple comparisons were used to test significance and differences between sample groups. Pearson's correlation coefficient was used to measure the linear correlation between two variables. Regression analysis was performed to determine the function to best fit our behavioral data and electrophysiologic data. Both linear and nonlinear (polynomial) equations were applied in the fitting. The goodness of fit was compared based on R-square.

**Reporting summary**. Further information on research design is available in the Nature Research Reporting Summary linked to this article.

## Data availability

Source data are provided with this paper. All the data for the figures (except anatomical images) are provided in the source data file. Anatomical images are available from the corresponding authors upon reasonable request. Analysis codes used in this study are available from GitHub (https://github.com/ZhongliUSC/Looming-Sound) and archived in Zenodo[62]. The Allen Brain Atlas (http://www.brain-map.org) was a reference for anatomical information in this study.

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

## Acknowledgements

This work was supported by grants from the US National Institutes of Health (R01DC008983 and RF1MH114112 to L.I.Z.; R01EY019049 to H.W.T.). We would like to thank Drs. Haifu Li and Feixue Liang for the generous help with experiments.

## Author contributions

Z.L. performed all the behavioural experiments. J.W. performed in vivo recording experiments. G.Z., J.H., X.W. and Z.L. performed anatomy, RNAscope and imaging experiments. B.Z. performed axon collateral analysis. L.I.Z. and H.W.T. supervised the study and wrote the manuscript.

## Competing interests

The authors declare no competing interests.
