## [Peer Review File · Nature Communications]

REVIEWER COMMENTS

Reviewer #1 (Remarks to the Author):

Zhong et al. have investigated the fascinating, relevant corticostriatal projection and their role in controlling defense behavior induced by looming acoustic threats. The authors compare the response of superior collicular and striatal neurons during looming acoustic sound presentation. The main finding is that the silencing of the SC with muscimol significantly impaired the crescendo-induced escape while freezing was not. On the other hand, the tail of the striatum TS is responsible for the crescendo-triggered freezing. Overall, the manuscript represents a first characterization of the essential role of the striatum in innate defense behavior.

The authors used a crescendo stimulus to analyze the behavioral response of the mouse. The stimuli consist of noise intensity that was increased from 20 to 70 dB SPL. Can the authors provide more info regarding the crescendo stimulus, such as the type of noise used, the frequency range of the noise, how the noise intensity was increased (for example: linear vs. exponential, speed), etc.... The same applies to the decrescendo stimulus.

The authors performed single-cell loose-patch recording in layer 5 of AC in awake head-fixed mice. Can the authors provide information about the onset of the ON and sustained response of the layer 5 neurons?

Does a frequency tuning characterize these neurons?

Can the authors speculate about the differences between layer 5 neurons (i.e., intrinsic bursting and regular spiking neurons)? Are the intrinsic bursting neurons showing sustained response while the regular spiking neurons show a phasic response to the crescendo stimulus?

The authors claimed that consistent with previous findings; they find that there is a spatially specific projection from AC to the intermediate part of the TS, where D2-type medium spiny neurons are clustered. However, that did not provide any evidence that the projection from AC preferentially targets the D2-type MSN neurons.

Can the authors provide more results about nature (for example, molecular markers) of cells labeled in the TS by using the anterograde transsynaptic methods?

Reviewer #2 (Remarks to the Author):

Summary:

This paper presents neural evidence from mice that strongly implicates the auditory cortex, superior colliculus, and striatum in differentially controlling freezing and escape responses to auditory stimuli. Specifically, the findings suggest that the auditory cortex controls freezing responses to crescendos via sustained neural firing that communicates with the tail of the striatum. The auditory cortex also modulates, but is not necessary for, escape responses controlled by the superior colliculus. Overall, the paper is well written and the analyses are easy to follow. The findings are clearly and concisely presented and form a well-rounded investigation that will be of significant interest to the field. My main suggestion for improvement is to build on the discussion of the design & findings, as well as some potential analyses.

Discussion points:

1. **Circuitry**: In mice, the majority of visual information from retinal ganglion cells is transmitted to the superior colliculus (SC), while auditory information is received by cochlear nucleus, and then transmitted to the inferior colliculus and then the medial geniculate nucleus of the thalamus (MGB). Hence, the superior colliculus is a source of primary visual - but not auditory - input.

a) As the authors mentioned (page 6), silencing the SC inhibits both escape and freezing responses to looming visual stimuli. In the present study, however, silencing the SC only inhibited escape behaviour. This is likely because the SC is the gateway for the majority of incoming visual information (thus generally disrupting visual processing & behavioural responses), whereas the MGB gates the majority of auditory input. Hence, the results of this study suggest that the SC takes a more downstream position for auditory than visual defense responses for specifically controlling escape - but not freezing - responses (likely by evoking a response in the periaqueductal gray; Evans et al., 2018, Nature).

b) Previous work by Joseph LeDoux has shown that simple auditory stimuli can elicit fearful responses even in the absence of the auditory cortex, due to the MGB's connections to other areas, such as the amygdala. More "complex" stimuli, however, appear to require the auditory cortex (LeDoux, 1998, Biological Psychiatry). In the present study, the sustained neural firing in the auditory cortex significantly increased freezing time and sped up escape. Hence, the auditory cortex likely performs crucial encoding of the crescendo - i.e., this is the key complexity that perhaps the MGB is unable to encode as effectively.

2. ****Stimuli****: How do the stimuli used in the present study compare to predatory/prey/conspicuous/environmental sounds relevant to mice?

a) From what I could find, the paper doesn't state what the frequency of the sounds were in Hz. It would be interesting to know whether the frequency of the sound falls within the range of short-range (social) vs. long-range (danger) mouse communication (Warren et al., 2020, Scientific Reports), or lower-range frequencies for movement of another animal.

b) What was the motivation for using bursts of stimulus repetition, rather than a single long and gradual crescendo/decrescendo? I'm not sure what sort of naturalistic threat such a stimulus might be similar to (perhaps a sort of panting sound of predator?). It doesn't seem to realistically reflect the sound of an approaching predator (there would be a single, not multiple, increasing sound). I understand that other studies in the visual domain have done a similar stimulus repetition (e.g., Shange et al., 2018, Nature Communications), but an explanation might still help the reader. It was mentioned on page 4 of the manuscript, where it is explained that behaviour adapts to repeated representations, and so only the first trial was used. Isn't this even more reason to have not done repeated presentations of the stimulus per trial?

c) Mice presumably perceive a looming visual stimulus as threatening because it indicates a flying predator overhead. The shadow itself, however, is not harmful. Auditory stimuli, however, can be a source of harm if presented at a painfully loud volume. In Extended Data Figure 1, it seems that stimuli at 90dB and 100dB, as well as the 0 s rise stimuli (which would be louder for longer), produce primarily escape behaviour and less freezing. If the mice habituate to these stimuli over trials, then presumably the stimuli don't cause harm and the response is indeed to a signal or threat.

Analytic points:

1. The findings presented in Fig 3.g-i demonstrate that a potential trade-off mechanism between the SC and TS for when freezing turns into escape. There are two elements that could greatly enhance this finding:

a) Does the escape onset time significantly correlate with the diminishing of TS firing?

b) Were there recordings also taken from the auditory cortex? If the freezing responses in TS are indeed modulated by the sustained firing of the auditory cortex, we might expect the firing in the auditory cortex to diminish alongside a reduction in TS firing (and, in turn, less freezing / more

movement).

Minor edits:

1. The paper doesn't appear to be split into introduction/method/results/discussion sections. It can be deduced (e.g., results start at the end of page 3, discussion starts at the end of page 8), but would be easier if the authors indicate the sections using titles.

2. I found it difficult to understand the order of the conditions, as well as which mice were put into which condition (e.g., in Fig. 2f-g, are the datapoints for Ctrl and LED from the *same* 7 mice in a repeated-measures design or were there 7 mice in two different groups?). It should be stated explicitly throughout the results section how many mice went into each analysis, and to which condition(s) they were assigned.

3. It would be beneficial to provide videos of the experiment (if available), as well as to provide the refined data and analytic code on a publicly-accessible repository (e.g., Open Science Framework, GitHub, FigShare, etc.)

Your sincerely,

Jessica McFadyen

Reviewer #3 (Remarks to the Author):

The paper entitled "Corticostriatal Control of Defense Behavior Induced by Looming Acoustic Threats" showed that the primary auditory cortex is required for looming stimuli induced freezing and flight behaviors, and they identified two pathways downstream of the primary auditory cortex that differentially mediate these freezing and flight behaviors. Overall the study is well designed and organized, and the data convincingly supports the conclusions. I believe this study will advance our current understanding in both neural mechanisms underlying innate behaviors and the functions of the tail of dorsal striatum.

I only have a couple minor comments. Given the maturity of this study at its current version, if further experiments are not feasible during this pandemic situation, discussions on these issues are also fine.

1. To better understand when the auditory pathway diverges to regulate this freezing and escape behaviors, it would be interesting to know whether the same or different populations of auditory cortical neurons project to the tail striatum and superior colliculus.

2. The authors showed that when the peak intensity of looming stimuli increased, mice tended to change from freeze-escape to escape only (Extended Data Fig. 1). Do the stimulus-evoked responses in TS and SC (Fig. 3 h & i) change when the peak intensity of the stimulus increased? Will this change of responses (if there is) explain the change in behavior?

REVIEWER COMMENTS

Reviewer #1 (Remarks to the Author):

Zhong et al. have investigated the fascinating, relevant corticostriatal projection and their role in controlling defense behavior induced by looming acoustic threats. The authors compare the response of superior collicular and striatal neurons during looming acoustic sound presentation. The main finding is that the silencing of the SC with muscimol significantly impaired the crescendo-induced escape while freezing was not. On the other hand, the tail of the striatum TS is responsible for the crescendo-triggered freezing. Overall, the manuscript represents a first characterization of the essential role of the striatum in innate defense behavior.

The authors used a crescendo stimulus to analyze the behavioral response of the mouse. The stimuli consist of noise intensity that was increased from 20 to 70 dB SPL. Can the authors provide more info regarding the crescendo stimulus, such as the type of noise used, the frequency range of the noise, how the noise intensity was increased (for example: linear vs. exponential, speed), etc.... The same applies to the decrescendo stimulus.

The authors performed single-cell loose-patch recording in layer 5 of AC in awake head-fixed mice. Can the authors provide information about the onset of the ON and sustained response of the layer 5 neurons? Does a frequency tuning characterize these neurons? Can the authors speculate about the differences between layer 5 neurons (i.e., intrinsic bursting and regular spiking neurons)? Are the intrinsic bursting neurons showing sustained response while the regular spiking neurons show a phasic response to the crescendo stimulus?

The authors claimed that consistent with previous findings; they find that there is a spatially specific projection from AC to the intermediate part of the TS, where D2-type medium spiny neurons are clustered. However, that did not provide any evidence that the projection from AC preferentially targets the D2-type MSN neurons. Can the authors provide more results about nature (for example, molecular markers) of cells labeled in the TS by using the anterograde transsynaptic methods?

Reviewer #2 (Remarks to the Author):

Summary:

This paper presents neural evidence from mice that strongly implicates the auditory cortex, superior colliculus, and striatum in differentially controlling freezing and escape responses to auditory stimuli. Specifically, the findings suggest that the auditory cortex controls freezing responses to crescendos via sustained neural firing that communicates with the tail of the striatum. The auditory cortex also modulates, but is not necessary for, escape responses controlled by the superior colliculus. Overall, the paper is well written and the analyses are easy to follow. The findings are clearly and concisely presented and form a well-rounded

investigation that will be of significant interest to the field. My main suggestion for improvement is to build on the discussion of the design & findings, as well as some potential analyses.

Discussion points:

1. **Circuitry**: In mice, the majority of visual information from retinal ganglion cells is transmitted to the superior colliculus (SC), while auditory information is received by cochlear nucleus, and then transmitted to the inferior colliculus and then the medial geniculate nucleus of the thalamus (MGB). Hence, the superior colliculus is a source of primary visual - but not auditory - input.

a) As the authors mentioned (page 6), silencing the SC inhibits both escape and freezing responses to looming visual stimuli. In the present study, however, silencing the SC only inhibited escape behaviour. This is likely because the SC is the gateway for the majority of incoming visual information (thus generally disrupting visual processing & behavioural responses), whereas the MGB gates the majority of auditory input. Hence, the results of this study suggest that the SC takes a more downstream position for auditory than visual defense responses for specifically controlling escape - but not freezing - responses (likely by evoking a response in the periaqueductal gray; Evans et al., 2018, Nature).

b) Previous work by Joseph LeDoux has shown that simple auditory stimuli can elicit fearful responses even in the absence of the auditory cortex, due to the MGB's connections to other areas, such as the amygdala. More "complex" stimuli, however, appear to require the auditory cortex (LeDoux, 1998, Biological Psychiatry). In the present study, the sustained neural firing in the auditory cortex significantly increased freezing time and sped up escape. Hence, the auditory cortex likely performs crucial encoding of the crescendo - i.e., this is the key complexity that perhaps the MGB is unable to encode as effectively.

2. **Stimuli**: How do the stimuli used in the present study compare to predatory/prey/conspicuous/environmental sounds relevant to mice?

a) From what I could find, the paper doesn't state what the frequency of the sounds were in Hz. It would be interesting to know whether the frequency of the sound falls within the range of short-range (social) vs. long-range (danger) mouse communication (Warren et al., 2020, Scientific Reports), or lower-range frequencies for movement of another animal.

b) What was the motivation for using bursts of stimulus repetition, rather than a single long and gradual crescendo/decrecendo? I'm not sure what sort of naturalistic threat such a stimulus might be similar to (perhaps a sort of panting sound of predator?). It doesn't seem to realistically reflect the sound of an approaching predator (there would be a single, not multiple, increasing sound). I understand that other studies in the visual domain have done a similar stimulus repetition (e.g., Shange et al., 2018, Nature Communications), but an explanation might still help the reader. It was mentioned on page 4 of the manuscript, where it is explained that behaviour adapts to repeated representations, and so only the first trial was used. Isn't this even more reason to have not done repeated presentations of the stimulus per trial?

c) Mice presumably perceive a looming visual stimulus as threatening because it indicates a flying predator overhead. The shadow itself, however, is not harmful. Auditory stimuli, however, can be a source of harm if presented at a painfully loud volume. In Extended Data Figure 1, it seems that stimuli at 90dB and 100dB, as well as the 0 s rise stimuli (which would be louder for longer), produce primarily escape behaviour and less freezing. If the mice habituate to these stimuli over trials, then presumably the stimuli don't cause harm and the response is indeed to a signal or threat.

Analytic points:

1. The findings presented in Fig 3.g-i demonstrate that a potential trade-off mechanism between the SC and TS for when freezing turns into escape. There are two elements that could greatly enhance this finding:

a) Does the escape onset time significantly correlate with the diminishing of TS firing?

b) Were there recordings also taken from the auditory cortex? If the freezing responses in TS are indeed modulated by the sustained firing of the auditory cortex, we might expect the firing in the auditory cortex to diminish alongside a reduction in TS firing (and, in turn, less freezing / more movement).

Minor edits:

1. The paper doesn't appear to be split into introduction/method/results/discussion sections. It can be deduced (e.g., results start at the end of page 3, discussion starts at the end of page 8), but would be easier if the authors indicate the sections using titles.

2. I found it difficult to understand the order of the conditions, as well as which mice were put into which condition (e.g., in Fig. 2f-g, are the datapoints for Ctrl and LED from the *same* 7 mice in a repeated-measures design or were there 7 mice in two different groups?). It should be stated explicitly throughout the results section how many mice went into each analysis, and to which condition(s) they were assigned.

3. It would be beneficial to provide videos of the experiment (if available), as well as to provide the refined data and analytic code on a publicly-accessible repository (e.g., Open Science Framework, GitHub, FigShare, etc.)

Your sincerely,

Jessica McFadyen

Reviewer #3 (Remarks to the Author):

The paper entitled “Corticostriatal Control of Defense Behavior Induced by Looming Acoustic Threats” showed that the primary auditory cortex is required for looming stimuli induced freezing and flight behaviors, and they identified two pathways downstream of the primary auditory cortex that differentially mediate these freezing and flight behaviors. Overall the study is well designed and organized, and the data convincingly supports the conclusions. I believe this study will advance our current understanding in both neural mechanisms underlying innate behaviors and the functions of the tail of dorsal striatum.

I only have a couple minor comments. Given the maturity of this study at its current version, if further experiments are not feasible during this pandemic situation, discussions on these issues are also fine.

1. To better understand when the auditory pathway diverges to regulate this freezing and escape behaviors, it would be interesting to know whether the same or different populations of auditory cortical neurons project to the tail striatum and superior colliculus.
2. The authors showed that when the peak intensity of looming stimuli increased, mice tended to change from freeze-escape to escape only (Extended Data Fig. 1). Do the stimulus-evoked responses in TS and SC (Fig. 3 h & i) change when the peak intensity of the stimulus increased? Will this change of responses (if there is) explain the change in behavior?

Responses to reviewers:

We thank the reviewers for their time and effort in reviewing our manuscript “Corticostriatal Control of Defense Behavior Induced by Looming Acoustic Threats”. We appreciate the constructive suggestions and comments on our work. We have made every effort to address the concerns raised by the reviewers and follow their suggestions. Our point-by-point response (in blue) is provided below each of reviewers’ comments.

Responses to Reviewer #1:

...Overall, the manuscript represents a first characterization of the essential role of the striatum in innate defense behavior.

The authors used a crescendo stimulus to analyze the behavioral response of the mouse. The stimuli consist of noise intensity that was increased from 20 to 70 dB SPL. Can the authors provide more info regarding the crescendo stimulus, such as the type of noise used, the frequency range of the noise, how the noise intensity was increased (for example: linear vs. exponential, speed), etc.... The same applies to the decrescendo stimulus.

We have clarified the details of sound stimuli in the Methods (page 29). The crescendo/decrescendo stimulus we used was broadband white noise with the speaker calibrated for a frequency range of 1-60 kHz. Its intensity was linearly increased from 20 to 70 dB SPL (or linearly decreased from 70 to 20 dB SPL).

The authors performed single-cell loose-patch recording in layer 5 of AC in awake head-fixed mice. Can the authors provide information about the onset of the ON and sustained response of the layer 5 neurons? Does a frequency tuning characterize these neurons? Can the authors speculate about the differences between layer 5 neurons (i.e., intrinsic bursting and regular spiking neurons)? Are the intrinsic bursting neurons showing sustained response while the regular spiking neurons show a phasic response to the crescendo stimulus?

We thank the reviewer for the comments. We have categorized the recorded L5 neurons into regular spiking (RS) and intrinsic bursting (IB) subgroups according to their responses to best-frequency tones as we described previously (Sun et al., 2013), and compared their tonal receptive field properties. As shown in **Supplementary Fig. 5a-b**, both types of neurons are tuned for frequency with IB neurons more broadly tuned than RS neurons (**Supplementary Fig. 5c**). In addition, IB neurons have higher spontaneous and evoked firing rates and slightly faster tone responses than RS neurons (**Supplementary Fig. 5d-f**). Furthermore, we find that a majority of IB neurons are the sustained type while few of RS neurons belongs to this type (**Supplementary Fig. 5g**). Therefore, out of L5 corticofugal neurons the IB neurons contribute the most critically to the auditory looming-induced defense (page 14).

We have also provided information about the response onset latencies of ON and sustained neurons. Sustained and On responses exhibited similar onset latencies (sustained: 20.2 ± 0.6 ms, $n = 17$; On: 21.6 ± 0.4 ms, $n = 22$, $p = 0.68$, two-sided t test) (page 7).

The authors claimed that consistent with previous findings; they find that there is a spatially specific projection from AC to the intermediate part of the TS, where D2-type medium spiny neurons are clustered. However, that did not provide any evidence that the projection from AC preferentially targets the D2-type MSN neurons. Can the authors provide more results about

nature (for example, molecular markers) of cells labeled in the TS by using the anterograde transsynaptic methods?

We agree that it is important to provide more direct evidence for the cell type in TS innervated by AC axons. To this end, we have carried out RNAscope assay using probes to detect expression of D1 and D2 dopamine receptors. We injected AAV1-Cre in AC of Ai14 mice to transsynaptically label AC-recipient TS neurons. As shown in **Fig. 6f-h**, the tdTomato-labeled AC-recipient TS neurons were mainly located in a D2-enriched region. Among the labeled AC-recipient TS neurons, about 70% expressed D2 receptors while only about 20% expressed D1 receptors. These data demonstrate that AC axons preferentially target the D2-type of medium spiny neurons (MSNs) in TS.

Responses to Reviewer #2:

...The findings are clearly and concisely presented and form a well-rounded investigation that will be of significant interest to the field. My main suggestion for improvement is to build on the discussion of the design & findings, as well as some potential analyses.

Discussion points:

1. **Circuitry**: *In mice, the majority of visual information from retinal ganglion cells is transmitted to the superior colliculus (SC), while auditory information is received by cochlear nucleus, and then transmitted to the inferior colliculus and then the medial geniculate nucleus of the thalamus (MGB). Hence, the superior colliculus is a source of primary visual - but not auditory - input.*

a) As the authors mentioned (page 6), silencing the SC inhibits both escape and freezing responses to looming visual stimuli. In the present study, however, silencing the SC only inhibited escape behaviour. This is likely because the SC is the gateway for the majority of incoming visual information (thus generally disrupting visual processing & behavioural responses), whereas the MGB gates the majority of auditory input. Hence, the results of this study suggest that the SC takes a more downstream position for auditory than visual defense responses for specifically controlling escape - but not freezing - responses (likely by evoking a response in the periaqueductal gray; Evans et al., 2018, Nature).

Our main conclusion is that auditory information reaching auditory cortex is streamed into two corticofugal pathways, with one controlling freezing via TS and one controlling flight via SC. AC is required for both freezing and flight. Indeed, compared to visually induced defense, SC takes a more downstream position. We have added this point to the Discussion (page 13).

b) Previous work by Joseph LeDoux has shown that simple auditory stimuli can elicit fearful responses even in the absence of the auditory cortex, due to the MGB's connections to other areas, such as the amygdala. More "complex" stimuli, however, appear to require the auditory cortex (LeDoux, 1998, Biological Psychiatry). In the present study, the sustained neural firing in the auditory cortex significantly increased freezing time and sped up escape. Hence, the auditory cortex likely performs crucial encoding of the crescendo - i.e., this is the key complexity that perhaps the MGB is unable to encode as effectively.

This is a good point. We agree that the relatively high complexity of crescendo stimuli may require the auditory cortex to encode the threat signals. We have added this point in the Discussion (page 13) and cited references (LeDoux, 1998; Letzkus et al., 2011).

2. **Stimuli**: *How do the stimuli used in the present study compare to predatory/prey/conspecific/environmental sounds relevant to mice?*

a) From what I could find, the paper doesn't state what the frequency of the sounds were in Hz. It would be interesting to know whether the frequency of the sound falls within the range of short-range (social) vs. long-range (danger) mouse communication (Warren et al., 2020, Scientific Reports), or lower-range frequencies for movement of another animal.

We have clarified in the Methods that we used broadband white noise (page 29), which is commonly used as a general-purpose testing sound. It is very different from the communication sounds of mice (either short-range or long-range) in terms of frequency range and spectrotemporal structures. For example, mouse communication sounds consist of ultrasonic short frequency-modulated (FM) sweeps (Liu and Schreiner, 2007; Liu et al., 2003; Warren et al., 2020). We have added this point in the Discussion (page 12).

b) What was the motivation for using bursts of stimulus repetition, rather than a single long and gradual crescendo/decrescendo? I'm not sure what sort of naturalistic threat such a stimulus might be similar to (perhaps a sort of panting sound of predator?). It doesn't seem to realistically reflect the sound of an approaching predator (there would be a single, not multiple, increasing sound). I understand that other studies in the visual domain have done a similar stimulus repetition (e.g., Shange et al., 2018, Nature Communications), but an explanation might still help the reader. It was mentioned on page 4 of the manuscript, where it is explained that behaviour adapts to repeated representations, and so only the first trial was used. Isn't this even more reason to have not done repeated presentations of the stimulus per trial?

The reviewer has raised a good point. We agree that 10 crescendo stimuli may not realistically reflect the sound of an approaching predator, although we chose this repeated pattern of stimulation following previous visual studies. Nevertheless, in a separate set of experiments, we have applied only one single crescendo stimulus (70 dB SPL) with a long rise time (5s). This single stimulus was effective in inducing defense behavior (either freezing or escape) in 16 out of 21 animals (freezing duration: 3.2 ± 0.5 s; normalized top speed: 1.6 ± 0.2) with freezing-escape sequence observed in 4 mice. We have added the results of this experiment (page 14). However, the repeated pattern is more effective and robust ($p = 0.0071$, Fisher's exact test), which could be due to its higher effectiveness in attracting the animal's attention. We have included this point in the Discussion (page 13).

For the test of behavioral adaptation, each trial contains 10 crescendo stimuli.

c) Mice presumably perceive a looming visual stimulus as threatening because it indicates a flying predator overhead. The shadow itself, however, is not harmful. Auditory stimuli, however, can be a source of harm if presented at a painfully loud volume. In Extended Data Figure 1, it seems that stimuli at 90dB and 100dB, as well as the 0 s rise stimuli (which would be louder for longer), produce primarily escape behaviour and less freezing. If the mice habituate to these stimuli over trials, then presumably the stimuli don't cause harm and the response is indeed to a signal or threat.

Thanks to the reviewer for raising this issue. For rodents, a pulsed 90-100dB sound is usually not considered as harmful. 120 dB SPL pulsed sounds are routinely used to induce the auditory startle response, another type of defensive-like behavior, as to evaluate cognitive functions in rodents (Geyer and Swerdlow, 1998; Lauer et al., 2017; Pantoni et al., 2020). In this study, if escape behavior could be due to some physical pain caused by a loud sound, we should be able to see the escape under decrescendo stimuli as well since the animal would feel the "pain" even earlier than crescendo. However, in the same group of animals for which both crescendo and decrescendo stimuli of the same peak intensity were tested (**Fig. 1f-g**), decrescendo induced neither freezing nor escape. We think that this result justifies that the behaviors are in response to a threat signal cued by the temporal change in sound intensity.

Following the reviewer's suggestion, we have carefully checked the experiments in which we applied 90-100 dB crescendo stimuli for two trials (about 2 min inter-trial intervals). In the first trial, 5/9 animals showed freezing and 9/9 showed escape, while in the second trial only 3/9 animals showed freezing and 4/9 showed escape. This behavioral adaptation can be explained by the habituation of animals to sensory cues of potential threats but is difficult to be explained by responses to physical harm. We have added this point to the Discussion (page 12).

Analytic points:

1. The findings presented in Fig 3.g-i demonstrate that a potential trade-off mechanism between the SC and TS for when freezing turns into escape. There are two elements that could greatly enhance this finding:

a) Does the escape onset time significantly correlate with the diminishing of TS firing?

Thanks for this helpful comment. We have compared evoked responses of TS neurons over time and accumulative probability of escape latencies (**Fig. 5d**). Indeed, there was a strong negative correlation ($***p < 0.001$, $r = -0.92$, Pearson's correlation coefficient), suggesting that the decline of TS firing temporally correlates with the initiation of escape. Importantly, the initiation of escape is most likely to occur when TS neuron responses has rapidly declined (between 2 to 4 s). In addition, we also found a strong positive correlation between the decline of TS responses and the decline of probability of freezing ($***p < 0.001$, $r = 0.92$, Pearson's correlation coefficient) (**Fig. 5e**).

b) Were there recordings also taken from the auditory cortex? If the freezing responses in TS are indeed modulated by the sustained firing of the auditory cortex, we might expect the firing in the auditory cortex to diminish alongside a reduction in TS firing (and, in turn, less freezing / more movement).

We have compared the change of cortical neuron responses with the behavioral expression over time. There is a strong correlation between the decline of sustained neuron responses and the decline of probability of freezing ($***p < 0.001$, $r = 0.95$, Pearson's correlation coefficient) (**Fig. 5f**). On the other hand, transient On and Off responses were relatively stable over time (**Fig. 5g**). These results are in line with our behavioral data showing that sustained AC neurons contribute to crescendo-induced defense behaviors and suggest that TS neuron adaptation can be partially attributed to the adaptation of corticofugal neurons.

Minor edits:

1. The paper doesn't appear to be split into introduction/method/results/discussion sections. It can be deduced (e.g., results start at the end of page 3, discussion starts at the end of page 8), but would be easier if the authors indicate the sections using titles.

We split our manuscript into introduction/results/discussion sections, following the journal format. Also, we have added a subtitle to each section of the results.

*2. I found it difficult to understand the order of the conditions, as well as which mice were put into which condition (e.g., in Fig. 2f-g, are the datapoints for Ctrl and LED from the *same* 7*

mice in a repeated-measures design or were there 7 mice in two different groups?). It should be stated explicitly throughout the results section how many mice went into each analysis, and to which condition(s) they were assigned.

We agree that the x-axis title might be confusing. We have now clarified in the text that these were the same animals in two conditions (LED Off and LED On) in a repeated measures design (with sequence randomized). Data points for the same animal are connected with a line.

3. It would be beneficial to provide videos of the experiment (if available), as well as to provide the refined data and analytic code on a publicly-accessible repository (e.g., Open Science Framework, GitHub, FigShare, etc.)

We have now provided representative videos (Supplementary Video 1-5). The refined data and analytic code will also be provided with the publication of the paper.

Responses to Reviewer #3:

... Overall the study is well designed and organized, and the data convincingly supports the conclusions. I believe this study will advance our current understanding in both neural mechanisms underlying innate behaviors and the functions of the tail of dorsal striatum.

I only have a couple minor comments. Given the maturity of this study at its current version, if further experiments are not feasible during this pandemic situation, discussions on these issues are also fine.

1. To better understand when the auditory pathway diverges to regulate this freezing and escape behaviors, it would be interesting to know whether the same or different populations of auditory cortical neurons project to the tail striatum and superior colliculus.

The reviewer has raised a good point. To address this issue, we have performed dual retrograde dye labeling by injecting CTb of different colors in TS and SC respectively in the same animal (**Fig. 6a-c**). We found that TS- and SC-projecting AC neurons were largely separate populations, with little overlap between them. In addition, we analyzed axon collaterals of TS-projecting AC neurons and found that they project only very sparse axons to SC (**Fig. 6d-e**). This again confirms that AC-TS and AC-SC pathways are largely segregated.

2. The authors showed that when the peak intensity of looming stimuli increased, mice tended to change from freeze-escape to escape only (Extended Data Fig. 1). Do the stimulus-evoked responses in TS and SC (Fig. 3 h & i) change when the peak intensity of the stimulus increased? Will this change of responses (if there is) explain the change in behavior?

Thanks to the reviewer for these constructive comments. We have examined TS and SC neuron responses to crescendo stimuli of different intensities (50, 70 and 90 dB SPL). For TS neurons, we found that their responses increased from 50 to 70 dB SPL and then decreased from 70 to 90 dB SPL (**Fig. 5h**). This correlates with the initial increase and then decrease in the probability of freezing (**Fig. 5i**). In comparison, the evoked responses of SC neurons exhibited a consistent overall increase from 50 to 90 dB SPL (**Fig. 5j**), which correlates with the increase in the probability of escape (**Fig. 5k**). These results further support our conclusion that TS neuron activity underlies freezing and SC neuron activity underlies flight.

References

- Geyer, M.A., and Swerdlow, N.R. (1998). Measurement of Startle Response, Prepulse Inhibition, and Habituation. *Curr. Protoc. Neurosci.* 3, 8.7.1-8.7.15.
- Lauer, A.M., Behrens, D., and Klump, G. (2017). Acoustic startle modification as a tool for evaluating auditory function of the mouse: Progress, pitfalls, and potential. *Neurosci. Biobehav. Rev.* 77, 194–208.
- LeDoux, J. (1998). Fear and the brain: Where have we been, and where are we going? *Biol. Psychiatry* 44, 1229–1238.
- Letzkus, J.J., Wolff, S.B.E., Meyer, E.M.M., Tovote, P., Courtin, J., Herry, C., and Lüthi, A. (2011). A disinhibitory microcircuit for associative fear learning in the auditory cortex. *Nature* 480, 331–335.
- Liu, R.C., and Schreiner, C.E. (2007). Auditory cortical detection and discrimination correlates with communicative significance. *PLoS Biol.* 5, 1426–1439.
- Liu, R.C., Miller, K.D., Merzenich, M.M., and Schreiner, C.E. (2003). Acoustic variability and distinguishability among mouse ultrasound vocalizations. *J. Acoust. Soc. Am.* 114, 3412–3422.
- Pantoni, M.M., Herrera, G.M., Van Alstyne, K.R., and Anagnostaras, S.G. (2020). Quantifying the Acoustic Startle Response in Mice Using Standard Digital Video. *Front. Behav. Neurosci.* 14, 1–9.
- Warren, M.R., Clein, R.S., Spurrier, M.S., Roth, E.D., and Neunuebel, J.P. (2020). Ultrashort-range, high-frequency communication by female mice shapes social interactions. *Sci. Rep.* 10, 1–14.

REVIEWERS' COMMENTS

Reviewer #1 (Remarks to the Author):

The authors have carefully addressed all of my previous comments.

Reviewer #2 (Remarks to the Author):

The authors have done an excellent job at addressing all reviewer comments. I think the manuscript now presents even more of a breadth of analyses to support their claims, and all methodology is clear. The discussion section is also now very comprehensive.

Reviewer #3 (Remarks to the Author):

The authors have addressed all my concerns in the current revision.

REVIEWERS' COMMENTS

Reviewer #1 (Remarks to the Author):

The authors have carefully addressed all of my previous comments.

Reviewer #2 (Remarks to the Author):

The authors have done an excellent job at addressing all reviewer comments. I think the manuscript now presents even more of a breadth of analyses to support their claims, and all methodology is clear. The discussion section is also now very comprehensive.

Reviewer #3 (Remarks to the Author):

The authors have addressed all my concerns in the current revision.

Reply to reviewers

We are glad that all reviewers are now satisfied with our responses and don't have additional questions.